# A Consolidated Cross-Validation Algorithm for Support Vector Machines via Data Reduction

**Boxiang Wang**
Department of Statistics and Actuarial Science
University of Iowa
Iowa City, IA 52242, USA
`boxiang-wang@uiowa.edu`

**Archer Y. Yang**
Department of Mathematics and Statistics
McGill University
Montreal, QC H3A 0B9, Canada
`archer.yang@mcgill.ca`

## Abstract

We propose a consolidated cross-validation (CV) algorithm for training and tuning the support vector machines (SVM) on reproducing kernel Hilbert spaces. Our consolidated CV algorithm utilizes a recently proposed exact leave-one-out formula for the SVM and accelerates the SVM computation via a data reduction strategy. In addition, to compute the SVM with the bias term (intercept), which is not handled by the existing data reduction methods, we propose a novel two-stage consolidated CV algorithm. With numerical studies, we demonstrate that our algorithm is about an order of magnitude faster than the two mainstream SVM solvers, kernlab and LIBSVM, with almost the same accuracy.

## 1 Introduction

This paper concerns one of the most successful classifiers, the kernel support vector machine (SVM) (Cortes and Vapnik, 1995; Vapnik, 1995, 1998), which has been popularly used on structured data in the past two decades. The success of the SVM is mainly attributed to its appealing geometric interpretation, solid theoretical foundation, and high predictive power. To assess the predictive accuracy of the SVM, cross-validation (CV)(Wahba and Wold, 1975; Arlot and Celisse, 2010) is perhaps the most commonly used method in practice. In a $K$-fold CV procedure, the training data is randomly split into $K$ equal-sized groups. Based on data splitting, part of the data is used for training each competing model and the rest of the data is reserved for evaluating the prediction error. The model with the smallest CV error is finally elected. Typical choices of $K$ are 5, 10, or $n$ (the sample size), where $K = n$ yields the so-called leave-one-out cross-validation (LOOCV).

LOOCV is generally less used than ten-fold and five-fold CV, largely because of the two popular arguments: (1) high computational cost of LOOCV; (2) much larger variance than five-fold or ten-fold CV. We must point out that while the first argument is true in some sense, the second argument is not generally true about LOOCV. For instance, Kohavi (1995) and Hastie et al. (2009) argue that leave-one-out is almost unbiased, but it has high variance, leading to unreliable estimates. A series of revealing works, e.g., Burman (1989); Bengio and Grandvalet (2004); Molinaro et al. (2005); Zhang and Yang (2015), have shown that, both empirically and theoretically, for modeling procedures with low instability, LOOCV often has the smallest variability. For example, in the context of the kernel SVM, Wang and Zou (2021) provided convincing numerical examples to show that (1) LOOCV has almost no bias in estimating the generalization error; (2) LOOCV does not necessarily have higher variance than ten-fold and five-fold CV. Consequently LOOCV results in a smaller overall error when estimating the prediction error as compared with ten-fold and five-fold CV.

From the aforementioned arguments, we can see the only legitimate complaint of LOOCV would arise from its expensive computation, as a typical approach needs to fit the models $n$ times on the leave-one-out data before evaluating their performance with each of the sample removed, so the

computational cost is roughly $n$ times as large as the cost of a single fit on the full data. To mitigate the computational burden, Golub et al. (1979) proposed a shortcut formula of LOOCV for smoothing splines such that the whole computation time is of the same order of fitting a single model, and the shortcut formula later evolved into the generalized cross-validation (GCV) for ridge regression.

Nevertheless, for the kernel classifiers, how to efficiently compute the exact LOOCV is a long-standing open problem. The shortcut cross-validation formula has been long considered as a unique property of some linear smoothers, and many works such as the generalized approximated cross-validation (GACV) (Wahba et al., 1999) resorted to approximating LOOCV, while there is no theoretical guarantee that LOOCV can always be well approximated. To solve the exact (rather than approximated) LOOCV, until very recently, Wang and Zou (2022) successfully proposed a leave-one-out lemma extending the Golub-Heath-Wahba formula to the kernel classifiers. Specifically, they showed the exact LOOCV error can be obtained by slightly varying the class labels without literally leaving out some samples during the CV procedure. Since no sample is left out, all the folds of LOOCV are using the same complete data and thus redundant computational efforts can be saved to dramatically accelerate LOOCV. Based on the leave-one-out lemma, Wang and Zou (2022) unified the training and tuning of the SVM and developed a new `magicsvm` algorithm, which often runs a magnitude faster than the state-of-art SVM solvers, e.g., `kernlab` (Karatzoglou et al., 2004) and `LIBSVM` (Chang and Lin, 2011).

In this work, the main contribution is to propose a consolidated CV algorithm via data reduction. The data reduction method was first proposed by Ghaoui et al. (2010) for the lasso method (Tibshirani, 1996) and then extended to the SVM (Ogawa et al., 2013; Wang et al., 2014; Pan and Xu, 2018; Hong et al., 2019). The key renovation of our proposal is to reduce all the cross-validated data in a consolidated manner, thereby aiming to speedup the whole SVM procedure. Our method is fundamentally different from the existing methods which isolate the model training and tuning. Moreover, the existing data reduction methods cannot handle the SVM with the bias (intercept), which is essentially useful for achieving high prediction accuracy. To handle the SVM with the bias, we propose a novel *two-stage consolidated CV*; such an extension is highly non-trivial.

We implement the consolidate CV in a `ccvsvm` algorithm. Simulations and nine benchmark data are used to demonstrate the superior performance of `ccvsvm` To give a quick demonstration, our consolidated CV algorithm reduces the run time from more than 1.5 hours (by LIBSVM) to less than one minute, when performing the exact LOOCV for the kernel SVM on a data set arrhythmia.

The remainder of this paper is organized as follows. In Section 2, we discuss the exact leave-one-out lemma and then propose a consolidated CV algorithm via data reduction. Section 3 extends the consolidated CV to handle the general SVM problems with the bias. In Section 4, we demonstrate the computational advantages of fitting the kernel SVM using our proposed methods over the other competitors with simulations and real data applications. The paper is concluded in Section 5 with extensions through kernel approximations and discussions on future directions.

## 2 Methodology

### 2.1 SVM and the Exact Leave-One-Out Lemma

Since we need to work with the fundamentals of the SVM, we first review the SVM in this section.

We focus on binary classification. Let $L(u) = (1 - u)_+ = \max(1 - u, 0)$ be the *hinge loss*. Suppose there are $n$ training samples, $(\mathbf{x}_i, y_i)$, $i = 1, 2, \ldots, n$, where each $\mathbf{x}_i \in \mathbb{R}^p$ and $y_i = \pm 1$. The SVM can be formulated as a function estimation problem in a reproducing kernel Hilbert space (Wahba, 1990):

$$\hat{f}_l = \underset{f \in \mathcal{H}_K}{\operatorname{argmin}} \left[ \frac{1}{n} \sum_{i=1}^n (1 - y_i f(\mathbf{x}_i))_+ + \lambda_l \|f\|_{\mathcal{H}_K}^2 \right], \tag{1}$$

where $\lambda_l > 0$ is a tuning parameter chosen from a decreasing sequence $\lambda_1 > \lambda_2 > \ldots > \lambda_L$, $\mathcal{H}_K$, the RKHS, is generated by a bivariate kernel function $K : \mathcal{X} \times \mathcal{X} \to \mathbb{R}$, and the classifier $\hat{f}$ is thus dubbed *kernel SVM*. Throughout this paper, we consider the *universal kernel*, whose induced RKHS $\mathcal{H}_K$ is rich enough to yield arbitrarily accurate decision boundaries (Steinwart, 2001; Micchelli et al., 2006). A commonly used universal kernel is the radial kernel $K(\mathbf{x}_i, \mathbf{x}_j) = \exp(-\sigma \|\mathbf{x}_i - \mathbf{x}_j\|_2^2)$.

By the representer theorem (Wahba, 1990), problem (1) has a finite-dimensional solution:

$$\hat{\boldsymbol{\alpha}}_l = \operatorname*{argmin}_{\boldsymbol{\alpha} \in \mathbb{R}^n} \left[ \frac{1}{n} \sum_{i=1}^n (1 - y_i \mathbf{K}_i' \boldsymbol{\alpha})_+ + \lambda_l \boldsymbol{\alpha}' \mathbf{K} \boldsymbol{\alpha} \right], \tag{2}$$

where $\mathbf{K}$ is the $n \times n$ kernel matrix with $K_{ij} = K(\mathbf{x}_i, \mathbf{x}_j)$ and is assumed to be positive definite; $\mathbf{K}_i$ is its $i$th row. Thus problem (2) has a unique minimizer $\hat{f}(\mathbf{x}) = \sum_{i=1}^n \hat{\alpha}_i K(\mathbf{x}_i, \mathbf{x})$.

To tune the model, with the LOOCV procedure, the SVM is fitted on the training data with the $j$th sample opted out: for each $l = 1, 2, \ldots, L$ and each $j = 1, 2, \ldots, n$, let $\tilde{\boldsymbol{\alpha}}_l^{[-j]}$ be

$$\tilde{\boldsymbol{\alpha}}_l^{[-j]} = \operatorname*{argmin}_{\boldsymbol{\alpha} \in \mathbb{R}^{n-1}} \left[ \frac{1}{n} \sum_{i \neq j} \left( 1 - y_i (\mathbf{K}_i^{[-j]})' \boldsymbol{\alpha} \right)_+ + \lambda_l \boldsymbol{\alpha}' \mathbf{K}^{[-j]} \boldsymbol{\alpha} \right], \tag{3}$$

where $\mathbf{K}^{[-j]}$ is the leave-one-out kernel matrix induced by the training data without the $j$th sample. Problem (2) refers to *the complete data problem*, and problem (3) refers to *the LOOCV problem*.

The bottleneck of the LOOCV problem is mainly due to the computation involving $n$ different leave-one-out kernel matrices. To reduce the computational burden, this work is based on the *exact leave-one-out lemma* (Wang and Zou, 2022) for the kernel SVM, and the key idea is to obtain the exact LOOCV from the complete kernel matrix.

**Lemma 2.1.** *(Exact leave-one-out lemma) For a given $j$, let $\tilde{y}_i^{[j]} = y_i$ if $i \neq j$ and $\tilde{y}_j^{[j]} = 0$. Define*

$$\hat{\boldsymbol{\alpha}}_l^{[-j]} = \operatorname*{argmin}_{\boldsymbol{\alpha} \in \mathbb{R}^n} \left[ \frac{1}{n} \sum_{i=1}^n \left( 1 - \tilde{y}_i^{[j]} \mathbf{K}_i' \boldsymbol{\alpha} \right)_+ + \lambda_l \boldsymbol{\alpha}' \mathbf{K} \boldsymbol{\alpha} \right]. \tag{4}$$

*Then the solution of problem* (3) *can be obtained as*

$$\tilde{\boldsymbol{\alpha}}_l^{[-j]} = (\hat{\alpha}_{1,l}^{[-j]}, \ldots, \hat{\alpha}_{j-1,l}^{[-j]}, \hat{\alpha}_{j+1,l}^{[-j]}, \ldots, \hat{\alpha}_{n,l}^{[-j]})'.$$

Although problem (3) can be transformed into problem (4), the solutions of the two problems have different lengths. Lemma 2.1 indicates that $\hat{\boldsymbol{\alpha}}_l^{[-j]} = (\tilde{\alpha}_{1,l}^{[-j]}, \ldots, \tilde{\alpha}_{j-1,l}^{[-j]}, 0, \tilde{\alpha}_{j,l}^{[-j]}, \ldots, \tilde{\alpha}_{n-1,l}^{[-j]})'$, i.e. $\hat{\alpha}_{j,l}^{[-j]}$, the $j$th element of the solution $\hat{\boldsymbol{\alpha}}_l^{[-j]}$, is zero, and the solution of problem (3) can be retrieved by knocking off the $j$th element from $\hat{\boldsymbol{\alpha}}_l^{[-j]}$.

As a consequence of transforming problem (3) into problem (4), the same kernel matrix $\mathbf{K}$ is used in all the folds during LOOCV, rather than the leave-one-out matrices $\mathbf{K}^{[-j]}$, while slightly different responses are crafted for different $j$. By sharing the same kernel matrix, some redundant calculations can be saved and Wang and Zou (2022) developed the efficient algorithm *magicsvm*.

## 2.2 Consolidated CV via Data Reduction

On the basis of Lemma 2.1, we propose a data reduction strategy to accelerate the LOOCV computation of the kernel SVM, which is referred to as *consolidated CV*.

For notational convenience, the complete data problem (2) can be written as a special case of problem (4) with $j = 0$, i.e., $\hat{\boldsymbol{\alpha}}_l \equiv \hat{\boldsymbol{\alpha}}_l^{[-0]}$ and

$$\hat{\boldsymbol{\alpha}}_l^{[-0]} = \operatorname*{argmin}_{\boldsymbol{\alpha} \in \mathbb{R}^n} \left[ \frac{1}{n} \sum_{i=1}^n \left( 1 - \tilde{y}_i^{[0]} \mathbf{K}_i' \boldsymbol{\alpha} \right)_+ + \lambda_l \boldsymbol{\alpha}' \mathbf{K} \boldsymbol{\alpha} \right],$$

where we define $\tilde{y}_i^{[0]} = y_i$ for each $i = 1, 2, \ldots, n$. By solving problem (4) for all $j = 0, 1, \ldots, n$, we both train the SVM through the complete data problem (2) and tune it using LOOCV.

The idea of consolidated CV is motivated by the sparsity of the solution $\hat{\boldsymbol{\alpha}}_l^{[-j]}$ in problem (4). To see this, we check the optimality condition of problem (4) by taking the sub-differential of the objective with respect to each $\mathbf{K}_i' \boldsymbol{\alpha}$, for each $j = 0, 1, \ldots, n$:

$$0 \in \frac{1}{n} \tilde{y}_i^{[j]} \partial L \left( \tilde{y}_i^{[j]} \mathbf{K}_i' \hat{\boldsymbol{\alpha}}_l^{[-j]} \right) + 2 \lambda_l \hat{\alpha}_{i,l}^{[-j]}, \ \forall i = 1, \ldots, n,$$

where $\partial L(t)$ is the subgradient of the hinge loss function: $\partial L(t) = -1$, if $t < 1$; $\partial L(t) = 0$, if $t > 1$; and $\partial L(t) \in [-1, 0]$ if $t = 1$. It follows that

$$\hat{\alpha}_{i,l}^{[-j]} = \begin{cases} \dfrac{\tilde{y}_i^{[j]}}{2n\lambda_l}, & \text{if } \tilde{y}_i^{[j]}\mathbf{K}_i'\hat{\alpha}_l^{[-j]} < 1, \\ 0, & \text{if } \tilde{y}_i^{[j]}\mathbf{K}_i'\hat{\alpha}_l^{[-j]} > 1. \end{cases}$$

By translating $\tilde{y}_i^{[j]}$ back to $y_i$, we see

$$\hat{\alpha}_{i,l}^{[-j]} = \begin{cases} \dfrac{y_i}{2n\lambda_l}, & \text{if } y_i\mathbf{K}_i'\hat{\alpha}_l^{[-j]} < 1 \text{ and } i \neq j, \\ 0, & \text{if } y_i\mathbf{K}_i'\hat{\alpha}_l^{[-j]} > 1 \text{ or } i = j. \end{cases} \tag{5}$$

Expression (5) hints on a possible data reduction strategy: before invoking the actual calculation of $\hat{\alpha}_l^{[-j]}$, if we are advised that $y_i\mathbf{K}_i'\hat{\alpha}_l^{[-j]} > 1$ for some $i$, then we can directly set $\hat{\alpha}_{i,l}^{[-j]}$ to zero; likewise, if $y_i\mathbf{K}_i'\hat{\alpha}_l^{[-j]} < 1$ is given, then $\hat{\alpha}_{i,l}^{[-j]}$ must be $y_i/(2n\lambda_l)$ unless $i = j$. We can pre-determine the values of some coordinates and only need to focus on the calculation of the remaining ones. Hence the dimension of problem (4) can be reduced.

The key to performing the data reduction through expression (5) is to know whether $y_i\mathbf{K}_i'\hat{\alpha}_l^{[-j]} < 1$ or $> 1$ for some $i$ before $\hat{\alpha}_l^{[-j]}$ is actually computed. We present the following theorem.

**Theorem 2.2.** *For some $l > 1$, suppose we have solved*

$$\hat{\alpha}_{l-1} = \underset{\alpha \in \mathbb{R}^n}{\operatorname{argmin}} \left[ \frac{1}{n} \sum_{i=1}^{n} (1 - y_i\mathbf{K}_i'\alpha)_+ + \lambda_{l-1}\alpha'\mathbf{K}\alpha \right].$$

*For each $i = 1, 2, \ldots, n$, define*

$$a_{i,l}^+ = \frac{\lambda_{l-1} + \lambda_l}{2\lambda_l} y_i\mathbf{K}_i'\hat{\alpha}_{l-1} + \frac{\lambda_{l-1} - \lambda_l}{2\lambda_l} \sqrt{B}\sqrt{\hat{\alpha}_{l-1}'\mathbf{K}\hat{\alpha}_{l-1}} + \frac{B}{2n\lambda_l},$$

$$a_{i,l}^- = \frac{\lambda_{l-1} + \lambda_l}{2\lambda_l} y_i\mathbf{K}_i'\hat{\alpha}_{l-1} - \frac{\lambda_{l-1} - \lambda_l}{2\lambda_l} \sqrt{B}\sqrt{\hat{\alpha}_{l-1}'\mathbf{K}\hat{\alpha}_{l-1}} - \frac{B}{2n\lambda_l},$$

*where $B = \max_i K(\mathbf{x}_i, \mathbf{x}_i)$. Then for each $j = 0, 1, \ldots, n$, it holds*

$$a_{i,l}^- \leq y_i\mathbf{K}_i'\hat{\alpha}_l^{[-j]} \leq a_{i,l}^+, \ \forall i \neq j. \tag{6}$$

*Further, let $\mathcal{L} = \{i : a_{i,l}^+ < 1\}$ and $\mathcal{R} = \{i : a_{i,l}^- > 1\}$. Then the solution of problem* (4) *satisfies that*

$$\hat{\alpha}_{i,l}^{[-j]} = \begin{cases} \dfrac{\tilde{y}_i^{[j]}}{2n\lambda_l}, & \text{if } i \in \mathcal{L}; \\ 0, & \text{if } i \in \mathcal{R}. \end{cases}$$

In Theorem 2.2, for radial and Laplacian kernels, we can directly set $B = 1$; for some unbounded kernels such as polynomial kernels, we calculate $B = \max_{i \in \{1,2,\ldots,n\}} K(\mathbf{x}_i, \mathbf{x}_i)$ based on training data.

Note that Theorem 2.2 holds for $\hat{\alpha}_l^{[-j]}$, $\forall j = 0, 1, \ldots, n$. By utilizing knowledge of $\hat{\alpha}_{l-1}$, the solution of the complete data problem with the tuning parameter $\lambda_{l-1}$, we can pre-determining certain coordinates for both the complete data problem and all LOOCV problems with $\lambda_l$, i.e., $\hat{\alpha}_l^{[-j]}$ for all $j = 0, 1, \ldots, n$, through $\mathcal{L}$ and $\mathcal{R}$, thus performing data reduction in a *consolidated* fashion.

To solve problem (4), Theorem 2.2 implies that $\hat{\alpha}_{i,l}^{[-j]}$ for $i \in \mathcal{L}$ and $i \in \mathcal{R}$ can be pre-determined, so we only need to solve $\hat{\alpha}_{i,l}^{[-j]}$, for $i \in \mathcal{S}$ where $\mathcal{S} \equiv (\mathcal{L} \cup \mathcal{R})^C$. Denote by $\tau$ a one-to-one mapping from $\{1, 2, \ldots, n_s\}$ to $\mathcal{S}$, where $n_s$ is the cardinality of $\mathcal{S}$. Let $\Gamma$ be the $n \times n_s$ sub-matrix of $\mathbf{K}$ such that its $i$th column $\Gamma_i = \mathbf{K}_{\tau(i)}$. Let $\Sigma$ be the $n_s \times n_s$ matrix such that $\Sigma_{ij} = K_{\tau(i)\tau(j)}$.

---

**Algorithm 1** Consolidated cross-validation

---

**Input**: $\lambda_1 > \lambda_2 > \ldots > \lambda_L, \mathbf{K}, \mathbf{y}$.

  1: Obtain
$$\hat{\boldsymbol{\alpha}}_1 = \operatorname*{argmin}_{\boldsymbol{\alpha} \in \mathbb{R}^n} \frac{1}{n} \sum_{i=1}^{n} \left(1 - y_i \mathbf{K}_i' \boldsymbol{\alpha}\right)_+ + \lambda_1 \boldsymbol{\alpha}' \mathbf{K} \boldsymbol{\alpha}.$$

  2: **for** $l = 2, 3, \ldots, L$ **do**
  3:      Construct the sets $\mathcal{L}$ and $\mathcal{R}$ according to Theorem 2.2. Let $\mathcal{S} = (\mathcal{L} \cup \mathcal{R})^C$.
  4:      Construct the matrices $\mathbf{\Gamma}$ and $\mathbf{\Sigma}$.
  5:      **for** $j = 0, 1, \ldots, n$ **do**
  6:         **if** $j > 0$ and $\hat{\alpha}_{j,l} = 0$ **then**
  7:             Obtain $\hat{\boldsymbol{\alpha}}_l^{[-j]} = \hat{\boldsymbol{\alpha}}_l$.
  8:         **else**
  9:             Construct the vector $\bar{\mathbf{y}}^{[j]}$.
10:            Obtain $\hat{\boldsymbol{\eta}}_l^{[-j]}$ by solving problem (8). (If $j > 0$, initialize the algorithm by $\hat{\boldsymbol{\eta}}_l$.)
11:            Obtain $\hat{\boldsymbol{\alpha}}_l^{[-j]}$ from expression (7).
12:         **end if**
13:      **end for**
14: **end for**
**Output**: $\hat{\boldsymbol{\alpha}}_l, \hat{\boldsymbol{\alpha}}_l^{[-j]}$, for each $j = 1, 2, \ldots, n$ and $l = 1, 2, \ldots, L$.

---

For each $j = 0, 1, \ldots, n$, let $\bar{\mathbf{y}}^{[j]}$ be the $n$-vector with the $i$th element to be $\tilde{y}_i^{[j]}$ if $i \in \mathcal{S}$, and 0 if $i \notin \mathcal{S}$. The solution of problem (4) is obtained as

$$\hat{\alpha}_{i,l}^{[-j]} = \begin{cases} \dfrac{\tilde{y}_i^{[j]}}{2n\lambda_l}, & \text{if } i \in \mathcal{L}, \\ 0, & \text{if } i \in \mathcal{R}, \\ \hat{\eta}_{\tau^{-1}(i),l}^{[-j]}, & \text{if } i \in \mathcal{S}, \end{cases} \tag{7}$$

where $\hat{\eta}_{\tau^{-1}(i),l}^{[-j]}$ is the $\tau^{-1}(i)$th element of

$$\hat{\boldsymbol{\eta}}_l^{[-j]} = \operatorname*{argmin}_{\boldsymbol{\eta} \in \mathbb{R}^{n_s}} \left[ \frac{1}{n} \sum_{i=1}^{n} \left( 1 - \tilde{y}_i^{[j]} \mathbf{\Gamma}_i' \boldsymbol{\eta} - \frac{1}{2n\lambda_l} \tilde{y}_i^{[j]} \mathbf{K}_i' \bar{\mathbf{y}}^{[j]} \right)_+ + \frac{1}{n} \bar{\mathbf{y}}^{[j]'} \mathbf{\Gamma} \boldsymbol{\eta} + \lambda_l \boldsymbol{\eta}' \mathbf{\Sigma} \boldsymbol{\eta} \right]. \tag{8}$$

The dimension of problem (8) is $n_s$, which is lower than $n$ – the dimension of the original problem (4). The matrices $\mathbf{\Gamma}$ and $\mathbf{\Sigma}$ are the same for each $j = 0, 1, \ldots, n$. We shall introduce an optimization algorithm for solving problem (8) in the next section.

In addition, by utilizing a fact that an SVM solution is unchanged if non-support-vector data are left out, namely, $\hat{\alpha}_{j,l} = 0$ for some $j$ implies $\hat{\boldsymbol{\alpha}}_l^{[-j]} = \hat{\boldsymbol{\alpha}}_l$, we can directly obtain the $j$th LOOCV solution from the complete data problem without solving problem (8). We summarize the consolidated CV algorithm in Algorithm 1.

## 2.3 A Consolidated Algorithm for Solving Problem (8)

Due to Theorem 2.2, we can perform LOOCV by solving problem (8), a reduced-optimization problem, for each $j$. To overcome the computational challenge caused by non-smoothness of the hinge loss, we consider a smoothed loss,

$$L_\tau(u) = \begin{cases} 0 & u \geq 1 + \tau, \\ (u - (1 + \tau))^2/(4\tau) & 1 - \tau < u < 1 + \tau, \\ 1 - u & u \leq 1 - \tau, \end{cases}$$

for some small $\tau > 0$. One can show that $L_\tau$ has a Lipschitz continuous gradient, $|L_\tau'(t_1) - L_\tau'(t_2)| \leq \frac{1}{2\tau} |t_1 - t_2|, \forall t_1, t_2 \in \mathbb{R}$. Thus a smoothed surrogate of problem (8) is

$$\hat{\boldsymbol{\eta}}_{\tau,l}^{[-j]} = \operatorname*{argmin}_{\boldsymbol{\eta} \in \mathbb{R}^{n_s}} \left[ \frac{1}{n} \sum_{i=1}^{n} L^\tau \left( \tilde{y}_i^{[j]} \mathbf{\Gamma}_i' \boldsymbol{\eta} - \frac{1}{2n\lambda_l} \tilde{y}_i^{[j]} \mathbf{K}_i' \bar{\mathbf{y}}^{[j]} \right) + \frac{1}{n} \bar{\mathbf{y}}^{[j]'} \mathbf{\Gamma} \boldsymbol{\eta} + \lambda_l \boldsymbol{\eta}' \mathbf{\Sigma} \boldsymbol{\eta} \right]. \tag{9}$$

Problem (9) can be solved using the proximal gradient descent (PGD) algorithm (Parikh and Boyd, 2014). Specifically, the matrix inversion is computed first

$$\mathbf{P}^{-1} = \left( 2\lambda_l \boldsymbol{\Sigma} + \frac{1}{n\tau} \boldsymbol{\Gamma}'\boldsymbol{\Gamma} \right)^{-1}. \tag{10}$$

Then, for each $j = 0, 1, \ldots, n$, we update

$$\boldsymbol{\eta}^{[-j]} \leftarrow \boldsymbol{\eta}^{[-j]} - \mathbf{P}^{-1}\left( \boldsymbol{\Gamma}'\mathbf{z}^{(k)} + \frac{1}{n}\boldsymbol{\Gamma}'\bar{\mathbf{y}}^{[j]} + 2\lambda_l\boldsymbol{\Sigma}\boldsymbol{\eta}^{[-j]} \right) \tag{11}$$

until convergence, and then let $\hat{\boldsymbol{\eta}}_{\tau,l}^{[-j]} \leftarrow \boldsymbol{\eta}^{[-j]}$. We claim the above algorithm is consolidated since the same matrix inversion $\mathbf{P}^{-1}$ obtained from equation (10) can be used in equation (11) for all $j$ (all folds.) By saving huge computational efforts in inverting $n$ matrices, the consolidated CV algorithm is much more efficient than the standard CV implementation. We also include the warm-start, say, using $\hat{\boldsymbol{\eta}}_l$ to initialize $\hat{\boldsymbol{\eta}}_l^{[-j]}$ in problem (8), and Nesterov's acceleration to further boost the algorithm.

We just discussed the PGD algorithm for solving a smoothed SVM problem (9). Interestingly, the *exact* SVM solution based on problem (8) can be obtained by iteratively solving problem (9) with $\tau_1 > \tau_2 > \ldots$ where $\tau_1 = 1$ and $\tau_k = \tau_{k-1}/8$ for $k > 1$. The iteration is able to reach the exact solution of problem (8) in a finite number of steps, following a simple projection step. To conserve space, we omit details and refer interesting readers to Wang and Zou (2022).

## 3 Two-stage Consolidated CV for the General SVM Problems

The consolidated CV developed in Section 2 does not include the bias; nonetheless, the SVM without the bias may have lower prediction accuracy and its use is limited in certain applications. Although a regularized bias can be used by adding a constant feature to the data matrix, the standard practice of the SVM does not regularize the bias term. Thus our goal is to compute the SVM with the bias, namely, *the general SVM problems*. In this section, we extend the consolidated CV to handle the general SVM problems. Such an extension turns out to be non-trivial.

The general SVM problem is formulated as follows,

$$(\hat{\beta}_{0,l}, \hat{\boldsymbol{\alpha}}_l) = \underset{\beta_0 \in \mathbb{R}, \, \boldsymbol{\alpha} \in \mathbb{R}^n}{\operatorname{argmin}} \frac{1}{n} \sum_{i=1}^n \left[ 1 - y_i(\beta_0 + \mathbf{K}_i'\boldsymbol{\alpha}) \right]_+ + \lambda_l \boldsymbol{\alpha}'\mathbf{K}\boldsymbol{\alpha}, \tag{12}$$

and the corresponding LOOCV problems are, $j = 1, 2, \ldots, n$,

$$(\hat{\beta}_{0,l}^{[-j]}, \hat{\boldsymbol{\alpha}}_l^{[-j]}) = \underset{\beta_0 \in \mathbb{R}, \, \boldsymbol{\alpha} \in \mathbb{R}^n}{\operatorname{argmin}} \frac{1}{n} \sum_{i=1}^n \left[ 1 - \tilde{y}_i^{[j]}(\beta_0 + \mathbf{K}_i'\boldsymbol{\alpha}) \right]_+ + \lambda_l \boldsymbol{\alpha}'\mathbf{K}\boldsymbol{\alpha}. \tag{13}$$

For notational convenience, we let $\tilde{y}_i^{[0]} = y_i$ and let $(\hat{\beta}_{0,l}, \hat{\boldsymbol{\alpha}}_l) = (\hat{\beta}_{0,l}^{[-0]}, \hat{\boldsymbol{\alpha}}_l^{[-0]})$, so we extend problem (13) with $j = 0$ to include the complete data problem (12) as a special case.

The key difficulty of developing the consolidated CV procedure for the general SVM problems is that $|\hat{\beta}_{0,l} - \hat{\beta}_{0,l}^{[-j]}|$ is hard to bound. To this end, we propose a *two-stage consolidated CV procedure*, where we give a consolidated bound of $|\hat{\beta}_{0,l} - \hat{\beta}_{0,l}^{[-j]}|$ for all $j$ in the first stage.

For $l > 1$, suppose we have found the solutions of problems (12) and (13) with the tuning parameter $\lambda_{l-1}$. Denote these solutions by $(\hat{\beta}_{0,l-1}, \hat{\boldsymbol{\alpha}}_{l-1})$ and $(\hat{\beta}_{0,l-1}^{[-j]}, \hat{\boldsymbol{\alpha}}_{l-1}^{[-j]})$. In Lemma 3.1, for each $i$, we give a consolidated bound of $y_i\mathbf{K}_i'\hat{\boldsymbol{\alpha}}_l^{[-j]}$ for all $j = 0, 1, \ldots, n$ and $j \neq i$.

**Lemma 3.1.** *For each $i = 1, 2, \ldots, n$, define*

$$c_{i,l}^+ = \max_{\substack{j=0,1,\ldots,n \\ j \neq i}} \left\{ \frac{\lambda_{l-1} + \lambda_l}{2\lambda_l} y_i\mathbf{K}_i'\hat{\boldsymbol{\alpha}}_{l-1}^{[-j]} + \frac{\lambda_{l-1} - \lambda_l}{2\lambda_l} \sqrt{B} \sqrt{\hat{\boldsymbol{\alpha}}_{l-1}^{[-j]'}\mathbf{K}\hat{\boldsymbol{\alpha}}_{l-1}^{[-j]}} \right\},$$

$$c_{i,l}^- = \min_{\substack{j=0,1,\ldots,n \\ j \neq i}} \left\{ \frac{\lambda_{l-1} + \lambda_l}{2\lambda_l} y_i\mathbf{K}_i'\hat{\boldsymbol{\alpha}}_{l-1}^{[-j]} - \frac{\lambda_{l-1} - \lambda_l}{2\lambda_l} \sqrt{B} \sqrt{\hat{\boldsymbol{\alpha}}_{l-1}^{[-j]'}\mathbf{K}\hat{\boldsymbol{\alpha}}_{l-1}^{[-j]}} \right\}, \tag{14}$$

---

**Algorithm 2** Bi-section algorithm to find $\beta_{0,l}^+$ and $\beta_{0,l}^-$

---

**Input**: $\mathbf{y}, c_{i,l}^-, c_{i,l}^+, \epsilon = 10^{-7}$

1: Compute $B^+$ and $B^-$ as

$$B^+ = \max\left\{\max_{\{i:y_i=-1\}}\{c_{i,l}^+ - 1\}, \max_{\{i:y_i=1\}}\{1 - c_{i,l}^-\}\right\} + \epsilon,$$

$$B^- = \min\left\{\min_{\{i:y_i=-1\}}\{c_{i,l}^- - 1\}, \min_{\{i:y_i=1\}}\{1 - c_{i,l}^+\}\right\} - \epsilon.$$

2: Let $a^+ \leftarrow B^+, c^+ \leftarrow B^-, b^+ \leftarrow (a^+ + c^+)/2$.
3: **repeat**
4:    Compute $\psi^+(b^+)$.
5:    Let $a^+ \leftarrow b^+$ and $b^+ \leftarrow (b^+ + c^+)/2$ if $\psi^+(b^+) < 0$.
6:    Let $c^+ \leftarrow b^+$ and $b^+ \leftarrow (a^+ + b^+)/2$ if $\psi^+(b^+) \geq 0$.
7: **until** $|a^+ - c^+| < \epsilon$.
8: Let $\beta_{0,l}^+ \leftarrow a^+$.
9: Let $a^- \leftarrow B^+, c^- \leftarrow B^-, b^- \leftarrow (a^- + c^-)/2$.
10: **repeat**
11:    Compute $\psi^-(b^-)$.
12:    Let $a^- \leftarrow b^-$ and $b^- \leftarrow (b^- + c^-)/2$ if $\psi^-(b^-) \leq 0$.
13:    Let $c^- \leftarrow b^-$ and $b^- \leftarrow (a^- + b^-)/2$ if $\psi^-(b^-) > 0$.
14: **until** $|a^- - c^-| < \epsilon$.
15: Let $\beta_{0,l}^- \leftarrow c^-$.
**Output:** $\beta_{0,l}^+$ and $\beta_{0,l}^-$

---

where $B = \max_i K(\mathbf{x}_i, \mathbf{x}_i)$ and $B = 1$ for the radial kernel. Then for any $i = 1, \ldots, n$, it holds that

$$c_{i,l}^- \leq y_i \mathbf{K}_i' \hat{\boldsymbol{\alpha}}_l^{[-j]} \leq c_{i,l}^+, \ \forall j \neq i. \tag{15}$$

On the basis of the bounds of $y_i \mathbf{K}_i' \hat{\boldsymbol{\alpha}}_l^{[-j]}$ given in Lemma 3.1, we next present Lemma 3.2 and Lemma 3.3 to give bounds of $\hat{\beta}_{0,l}^{[-j]}$ that are consolidated for all $j = 0, 1, \ldots, n$.

**Lemma 3.2.** *With $c_{i,l}^-$ and $c_{i,l}^+$ from Lemma 3.1, for a given constant $b$, define $\mathcal{S}_1(b) = \{i : y_i b + c_{i,l}^+ < 1\}$ and $\mathcal{S}_2(b) = \{i : y_i b + c_{i,l}^- > 1\}$. Let $n_+(b) = \sum_{i \in (\mathcal{S}_1(b) \cup \mathcal{S}_2(b))^C} I(y_i = 1)$ and $n_-(b) = \sum_{i \in (\mathcal{S}_1(b) \cup \mathcal{S}_2(b))^C} I(y_i = -1)$. Define $\psi^+(b) = \sum_{i \in \mathcal{S}_1(b)} y_i + n_+(b) + 1$ and $\psi^-(b) = \sum_{i \in \mathcal{S}_1(b)} y_i - n_-(b) - 1$. Then we have*

*(1) both $\psi^+(b)$ and $\psi^-(b)$ are non-increasing in $b$;*

*(2) $\psi^+(b) < 0$ implies $b > \hat{\beta}_{0,l}^{[-j]}$ for all $j = 0, 1, \ldots, n$;*

*(3) $\psi^-(b) > 0$ implies $b < \hat{\beta}_{0,l}^{[-j]}$ for all $j = 0, 1, \ldots, n$.*

Following Lemma 3.2, we develop a bi-section algorithm in Algorithm 2 to give consolidated bounds for $\hat{\beta}_{0,l}^{[-j]}$ for all $j = 0, 1, \ldots, n$.

As shown in Lemma 3.3, Algorithm 2 yields consolidated bounds for $\hat{\beta}_{0,l}^{[-j]}$ for all $j = 0, 1, \ldots, n$.

**Lemma 3.3.** *Suppose the input of Algorithm 2, $c_{i,l}^-$ and $c_{i,l}^+$, satisfies inequality (15), then the output of Algorithm 2, $\beta_{0,l}^+$ and $\beta_{0,l}^-$, satisfies that*

$$\beta_{0,l}^- < \hat{\beta}_{0,l}^{[-j]} < \beta_{0,l}^+. \ \forall j = 0, 1, \ldots, n. \tag{16}$$

It immediately follows from Lemma 3.3 that

$$|\hat{\beta}_{0,l} - \hat{\beta}_{0,l}^{[-j]}| < \beta_{0,l}^+ - \beta_{0,l}^-, \tag{17}$$

for any $j$, achieving the goal of the first stage.

We have constructed the bounds in inequalities (15) and (16). However, these bounds are too loose to develop data reduction rules in practice. The loose bounds are mainly caused by the maximum and minimum operators that are involved in equations (14). To this end, in the second stage, we give refined bounds, which are presented below.

**Lemma 3.4.** *For each $i = 1, 2, \ldots, n$, define*

$$\tilde{c}_{i,l}^+ = \frac{\lambda_{l-1} + \lambda_l}{2\lambda_l} y_i \mathbf{K}_i' \hat{\boldsymbol{\alpha}}_{l-1} + \frac{\lambda_{l-1} - \lambda_l}{2\lambda_l} \sqrt{B} \sqrt{\hat{\boldsymbol{\alpha}}_{l-1}' \mathbf{K} \hat{\boldsymbol{\alpha}}_{l-1}} + \sqrt{\frac{B^2}{16n^2\lambda_l^2} + \frac{B(\beta_{0,l}^+ - \beta_{0,l}^-)}{2n\lambda_l}} + \frac{B}{4n\lambda_l},$$

$$\tilde{c}_{i,l}^- = \frac{\lambda_{l-1} + \lambda_l}{2\lambda_l} y_i \mathbf{K}_i' \hat{\boldsymbol{\alpha}}_{l-1} - \frac{\lambda_{l-1} - \lambda_l}{2\lambda_l} \sqrt{B} \sqrt{\hat{\boldsymbol{\alpha}}_{l-1}' \mathbf{K} \hat{\boldsymbol{\alpha}}_{l-1}} - \sqrt{\frac{B^2}{16n^2\lambda_l^2} + \frac{B(\beta_{0,l}^+ - \beta_{0,l}^-)}{2n\lambda_l}} - \frac{B}{4n\lambda_l},$$

*where $\beta_{0,l}^+$ and $\beta_{0,l}^-$ are produced by Algorithm 2. Then for any $j = 1, \ldots, n$, it holds that*

$$\tilde{c}_{i,l}^- \leq y_i \mathbf{K}_i' \hat{\boldsymbol{\alpha}}_l^{[-j]} \leq \tilde{c}_{i,l}^+, \ \forall j = 0, 1, \ldots, n, \ and \ j \neq i. \tag{18}$$

Hence by Lemmata 3.1 and 3.4, we have

$$\hat{c}_{i,l}^- \equiv \max\{c_{i,l}^-, \tilde{c}_{i,l}^-\} \leq y_i \mathbf{K}_i' \hat{\boldsymbol{\alpha}}_l^{[-j]} \leq \min\{c_{i,l}^+, \tilde{c}_{i,l}^+\} \equiv \hat{c}_{i,l}^+, \tag{19}$$

for any $j = 0, 1, \ldots, n$, and $j \neq i$. We then use $\max\{c_{i,l}^-, \tilde{c}_{i,l}^-\}$ and $\min\{c_{i,l}^+, \tilde{c}_{i,l}^+\}$ as the input in the bi-section algorithm to yield the output $\tilde{\beta}_{0,l}^+$ and $\tilde{\beta}_{0,l}^-$. By inequality (18) and Lemma 3.2, we have

$$\tilde{\beta}_{0,l}^- < \hat{\beta}_{0,l}^{[-j]} < \tilde{\beta}_{0,l}^+. \ \forall j = 0, 1, \ldots, n. \tag{20}$$

Therefore, we glean inequalities (18) and (20), which are the refined bounds of inequalities (15) and (16). Using the refined bounds, we now present the main theorem.

**Theorem 3.5.** *The solution of problem (12), $\hat{\boldsymbol{\alpha}}_l$, satisfies:*

$$\hat{\alpha}_{i,l} = \begin{cases} \dfrac{y_i}{2n\lambda_l}, & if \ i \in \tilde{\mathcal{L}}; \\ 0, & if \ i \in \tilde{\mathcal{R}}, \end{cases}$$

*and for any $j = 1, \ldots, n$, the solution of problem (13), $\hat{\boldsymbol{\alpha}}_l^{[-j]}$, satisfies:*

$$\hat{\alpha}_{i,l}^{[-j]} = \begin{cases} \dfrac{y_i}{2n\lambda_l}, & if \ i \in \tilde{\mathcal{L}} \ and \ i \neq j; \\ 0, & if \ i \in \tilde{\mathcal{R}} \ or \ i = j, \end{cases}$$

*where $\hat{c}_{i,l}^+$ and $\hat{c}_{i,l}^-$ are given in inequality (19) and*

$$\tilde{\mathcal{L}} = \left\{ i : y_i = 1 \ and \ \tilde{\beta}_{0,l}^+ + \hat{c}_{i,l}^+ < 1 \right\} \cup \left\{ i : y_i = -1 \ and \ -\tilde{\beta}_{0,l}^- + \hat{c}_{i,l}^+ < 1 \right\},$$

$$\tilde{\mathcal{R}} = \left\{ i : y_i = 1 \ and \ \tilde{\beta}_{0,l}^- + \hat{c}_{i,l}^- > 1 \right\} \cup \left\{ i : y_i = -1 \ and \ -\tilde{\beta}_{0,l}^+ + \hat{c}_{i,l}^- > 1 \right\}.$$

Thus by Theorem 3.5, problem (13) can be solved through some reduced-dimensional optimization problems, which are similar to problem (8) where the bias is excluded. Therefore, we can follow the discussions in Section 2.3 to employ the same PGD algorithm and the exact smoothing technique to obtain the exact solution for problem (13). Details of the algorithm are omitted to conserve space.

Table 1: Run time (in second), objective value, and test error of four kernel SVM solvers under mixture Gaussian distributed data with $p = \{20, 200\}$, and $n = \{200, 400, 800, 1600\}$. The test error is assessed on independently generated test data. The numbers are the average quantities over 50 independent runs and the standard errors are presented in parentheses.

| $p$ | $n$ | method | time (s) | objective | test error | method | time (s) | objective | test error |
|---|---|---|---|---|---|---|---|---|---|
| 20 | 200 | ccvsvm | 5.1 | 0.814 (.005) | 0.351 (.007) | kernlab | 73.4 | 0.814 (.005) | 0.351 (.007) |
| | | magicsvm | 7.7 | 0.814 (.005) | 0.351 (.007) | LIBSVM | 144.4 | 0.828 (.014) | 0.351 (.007) |
| | 400 | ccvsvm | 44.2 | 0.827 (.003) | 0.332 (.005) | kernlab | 334.3 | 0.827 (.003) | 0.332 (.005) |
| | | magicsvm | 87.8 | 0.827 (.003) | 0.332 (.005) | LIBSVM | 879.7 | 0.827 (.003) | 0.332 (.005) |
| | 800 | ccvsvm | 446.8 | 0.846 (.002) | 0.309 (.002) | kernlab | 2220.2 | 0.846 (.002) | 0.309 (.002) |
| | | magicsvm | 847.3 | 0.846 (.002) | 0.309 (.002) | LIBSVM | 6519.7 | 0.846 (.002) | 0.310 (.002) |
| | 1600 | ccvsvm | 3829.5 | 0.853 (.001) | 0.297 (.001) | kernlab | 25530.5 | 0.853 (.001) | 0.297 (.001) |
| | | magicsvm | 7024.1 | 0.853 (.001) | 0.297 (.001) | LIBSVM | 63886.1 | 0.853 (.001) | 0.297 (.001) |
| 200 | 200 | ccvsvm | 6.8 | 0.780 (.006) | 0.337 (.015) | kernlab | 337.6 | 0.780 (.006) | 0.339 (.015) |
| | | magicsvm | 12.9 | 0.780 (.006) | 0.337 (.015) | LIBSVM | 932.5 | 0.780 (.006) | 0.342 (.015) |
| | 400 | ccvsvm | 66.0 | 0.794 (.003) | 0.366 (.015) | kernlab | 2304.1 | 0.794 (.003) | 0.366 (.015) |
| | | magicsvm | 150.1 | 0.794 (.003) | 0.366 (.015) | LIBSVM | 6641.9 | 0.794 (.003) | 0.368 (.015) |
| | 800 | ccvsvm | 530.4 | 0.811 (.001) | 0.346 (.015) | kernlab | 36771.4 | 0.811 (.001) | 0.346 (.015) |
| | | magicsvm | 996.1 | 0.811 (.001) | 0.346 (.015) | LIBSVM | 109365.5 | 0.811 (.001) | 0.346 (.015) |
| | 1600 | ccvsvm | 5489.2 | 0.821 (.001) | 0.322 (.013) | kernlab | 461245.7 | 0.821 (.001) | 0.322 (.013) |
| | | magicsvm | 10803.9 | 0.821 (.001) | 0.322 (.013) | LIBSVM | 1436416.1 | 0.821 (.001) | 0.322 (.013) |

# 4 Numerical Studies

In this section, we demonstrate the computational advantages of fitting the kernel SVM using `ccvsvm` over the three other competitors, `magicsvm`, `kernlab`, and `LIBSVM`, with simulations and real data.

## 4.1 Simulations

A commonly used simulation data from mixture Gaussian distributions (Hastie et al., 2009) is used. We generate mean vectors $\boldsymbol{\mu}_{k_+}$ from $N(\boldsymbol{\mu}_+, \mathbf{I}_p)$ where $k = 1, 2, \ldots, 10$ in which $\boldsymbol{\mu}_+ = (1, 1, \ldots, 1, 0, 0, \ldots, 0)$ with half of the coordinates to be zero. Each positive-class training sample is independently generated from $N(\boldsymbol{\mu}_{k_+}, 3^2)$ where $k$ is drawn from the discrete uniform distribution on $\{1, 2, \ldots, 10\}$. Using the same procedure, we obtain the negative-class training data from $N(\boldsymbol{\mu}_{k_-}, 3^2)$ where $k$ is also uniform on $\{1, 2, \ldots, 10\}$ and $\boldsymbol{\mu}_- = (0, 0, \ldots, 0, 1, 1, \ldots, 1)$. For each combination of the feature dimension $p = 20$ and $200$ and the sample size $n = 200, 400, 800,$ and $1600$, we fit the kernel SVM using the four kernel SVM solvers, `ccvsvm`, `magicsvm`, `kernlab`, and `LIBSVM`, to compute the entire solution paths at a sequence of 50 tuning parameters, uniformly distributed on the logarithm scale between $e^{-6}$ and $e^6$. The radial kernel is used and the bandwidth is the default option of `kernlab`, which generally performs well. We compared the run time, objective function value, and test error of the four solvers, where the run time includes the whole computation process including training and tuning the model. The objective function value is computed from equation (2). Test error is assessed on $10,000$ test samples which are independently generated from the same distribution. Computations were conducted on an Intel(R) Xeon(R) Gold 6230 CPU @ 2.10 GHz.

Table 1 shows that, to reach the same objective value and the test error, our `ccvsvm` algorithm is roughly twice as fast as `magicsvm`, and it is about an order of magnitude faster than `kernlab` and `LIBSVM`. In addition, we observe that `kernlab` and `LIBSVM` significantly slow down as $p$ increases, e.g., `LIBSVM` is about 20 times slower when $p$ grows from 20 to 200, whereas the speed of `ccvsvm` and `magicsvm` is quite insensitive to the change of dimensions. Remarkably, for $p = 200$ and $n = 1600$, our `ccvsvm` algorithm finishes training and tuning the SVM using LOOCV within two hours, while with the same accuracy, `LIBSVM` spends about 400 hours, lasting over 16 days.

We exemplify the effect of data reduction using the simulation data with $n = 800$ and $p = 20$ and profile the execution time for training and tuning the SVM with $\lambda = 0.1$. We observe `magicsvm` took only 0.12 seconds for matrix inversions and 11.17 seconds for LOOCV through problem (4), whereas `ccvsvm` spent 0.03 seconds on matrix inversions and 5.81 seconds on LOOCV via problem (8). The advantage of `ccvsvm` over `magicsvm` is mainly attributed to the reduced dimension of problem (8) compared with problem (4).

Table 2: Run time (in second) of four SVM solvers for benchmark data, averaged over 50 runs.

| data | $n$ | $p$ | ccvsvm | magicsvm | kernlab | LIBSVM |
|------|-----|-----|--------|----------|---------|--------|
| arrhythmia | 452 | 191 | 48.076 | 113.099 | 1554.579 | 5061.881 |
| australian | 690 | 14 | 202.863 | 412.323 | 902.463 | 2178.644 |
| chess | 3196 | 37 | 21768.612 | 38942.348 | > 240 hours | > 240 hours |
| heart | 270 | 13 | 8.464 | 16.466 | 89.373 | 168.477 |
| leuk | 72 | 7218 | 0.464 | 0.828 | 1548.724 | 4811.612 |
| malaria | 71 | 22283 | 0.504 | 0.819 | 4804.442 | 13835.143 |
| musk | 476 | 166 | 62.169 | 127.656 | 1563.262 | 4778.779 |
| sonar | 208 | 60 | 4.736 | 7.033 | 98.080 | 221.505 |
| valley | 606 | 100 | 149.034 | 311.147 | 2230.010 | 6428.014 |

## 4.2 Benchmark Data Applications

We test the performance on benchmark data applications. We study nine commonly used real data applications from the UCI machine learning repository (Dua and Graff, 2017). The sample sizes range from 208 to 3, 196. Two high-dimensional data sets with the number of features $p = 7, 218$ and 22, 283 are included. Each data set is split into a training set and a test set with the ratio 9 : 1. The kernel SVM is trained and tuned by the four solvers on the training set, and the test error is assessed on the test set. We adopt the training-test split-ratio 9 : 1 because we aim to assign most of the samples to the training set and the computation time can be evaluated using relatively large data.

Table 2 exhibits the timing comparisons, where we discover our ccvsvm algorithm is clearly the fastest. It is about as twice as fast as magicsvm and significantly faster than kernlab and LIBSVM. Especially for the two high-dimensional examples, magicsvm is thousands or even tens of thousands faster than kernlab and LIBSVM, and ccvsvm further cuts the run time of magicsvm into half. Similar to the simulations, all the four kernel SVM solvers deliver almost the same objective values and test errors on the real data applications; for sake of space limit, the accuracy results are omitted.

## 5 Discussions and Extensions

In this work, we have introduced a consolidated CV procedure and developed an algorithm called ccvsvm for the kernel SVM, which is one of the most successful classifiers. Our work is built on the recently proposed leave-one-out lemma and the magicsvm algorithm: the ccvsvm algorithm can even double the speed of magicsvm, which has already shown remarkable computational advantages over the mainstream SVM solvers, kernlab and LIBSVM.

**Scaling ccvsvm to large data sets.** For large-scale data, we suggest incorporating kernel approximation into the existing consolidated CV algorithm. Specifically, random features (Rahimi and Recht, 2007) or Nyström subsampling (Rudi et al., 2015) can be applied in the exact leave-one-out formula of the SVM to find a low-cost approximation of the kernel matrix. Integrating these approximation techniques into our methods can further improve the numerical performance. These strategies can also improve generalization performances as they induce a form of implicit *computational regularization*. In the supplemental materials (Section C), we develop consolidated CV methods with kernel approximation, essentially converting the original consolidated kernel SVM to a consolidated linear SVM, which then can be efficiently solved by the proposed ccvsvm algorithm. To give a quick demonstration, we consider the simulation example in Section 4.1 with $p = 20$ and increase $n$ to be 5, 000, 000. Averaged by 50 runs, the SVM with random features can be rapidly trained and tuned in 831 seconds, giving test error 0.286 which is close to Bayes error, 0.260. The computation time is only 15.7 seconds when $n = 100, 000$. However, when $n$ is 800, the test error of the SVM with random features is 0.351, which is well above 0.309, the test error of our exact kernel SVM solver given in Table 1. We leave full investigations of this strategy to future works.

**Limitation.** The proposed method is only for LOOCV and SVM since it utilizes the special structure of support vectors. However, in future works, it is interesting to explore if the consolidated CV can be generalized to other $K$-fold CV or the hold-out validation more broadly. It is also interesting to extended the idea of consolidated CV to solve the solution paths of other computationally expensive machine learning methods such as support vector regression and kernel quantile regression.

**Societal impact.** This work does not present any foreseeable societal consequence.

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
