# Supplemental Materials: A Consolidated Cross-Validation Algorithm for Support Vector Machines via Data Reduction

## A  Technical Proofs

### A.1  Some details of Lemma 2.1

Since matrix $\mathbf{K}$ is positive definite, we can transform $\boldsymbol{\alpha}$ in (4) using $\boldsymbol{\theta} = \mathbf{K}\boldsymbol{\alpha}$, therefore $\boldsymbol{\alpha} = \mathbf{K}^{-1}\boldsymbol{\theta}$. We can then rewrite the minimization problem (4) with respect to $\boldsymbol{\theta}$:

$$\hat{\boldsymbol{\theta}}_l^{[-j]} = \operatorname*{argmin}_{\boldsymbol{\theta} \in \mathbb{R}^n} \left[ \frac{1}{n} \sum_{i=1}^n \left( 1 - \tilde{y}_i^{[j]} \theta_i \right)_+ + \lambda_l \boldsymbol{\theta}' \mathbf{K}^{-1} \boldsymbol{\theta} \right]. \tag{21}$$

Once $\hat{\boldsymbol{\theta}}_l^{[-j]}$ is obtained, we can compute $\hat{\boldsymbol{\alpha}}_l^{[-j]} = \mathbf{K}^{-1}\hat{\boldsymbol{\theta}}_l^{[-j]}$. The optimality condition of problem (21) with respect to $\boldsymbol{\theta}$ is

$$0 \in \frac{1}{n} \tilde{y}_i^{[j]} \partial L \left( \tilde{y}_i^{[j]} \hat{\theta}_{i,l}^{[-j]} \right) + 2\lambda_l (\mathbf{K}^{-1} \hat{\boldsymbol{\theta}}_l^{[-j]})_i, \forall i = 1, \ldots, n,$$

which yields the optimality condition (2.2) after applying $\hat{\boldsymbol{\alpha}}_l^{[-j]} = \mathbf{K}^{-1}\hat{\boldsymbol{\theta}}_l^{[-j]}$.

### A.2  Proof of Theorem 2.2

We first prove inequality (6) for $j = 0$, namely, the bound for $y_i \mathbf{K}_i' \hat{\boldsymbol{\alpha}}_l$. When $j = 0$, problem (4) reduces to problem (2), whose sub-gradient optimality condition (aka, Karush–Kuhn–Tucker condition) with respect to each $\mathbf{K}_i' \boldsymbol{\alpha}_l$ gives

$$0 \in \frac{1}{n} y_i \partial L(\mathbf{K}_i' \hat{\boldsymbol{\alpha}}_l) + 2\lambda_l \hat{\alpha}_{i,l}, \ \forall i, \tag{22}$$

where $\partial L$ is the subgradient of the hinge loss. For any $\boldsymbol{\alpha}$, define $g(\boldsymbol{\alpha}) = \frac{1}{n} \sum_{i=1}^n [1 - y_i \mathbf{K}_i' \boldsymbol{\alpha}]_+$. The convexity of $g$ implies

$$g(\hat{\boldsymbol{\alpha}}_{l-1}) \geq g(\hat{\boldsymbol{\alpha}}_l) + \frac{1}{n} \sum_{i=1}^n y_i v_i \mathbf{K}_i' (\hat{\boldsymbol{\alpha}}_l - \hat{\boldsymbol{\alpha}}_{l-1}), \tag{23}$$

for any $v_i \in \partial L(y_i \mathbf{K}_i' \hat{\boldsymbol{\alpha}}_l)$. Expression (22) indicates $v_i = -2\lambda_l n y_i \hat{\alpha}_{i,l} \in \partial L(y_i \mathbf{K}_i' \hat{\boldsymbol{\alpha}}_l)$. By using $v_i = -2\lambda_l n y_i \hat{\alpha}_{i,l}$ in expression (23) we see

$$g(\hat{\boldsymbol{\alpha}}_{l-1}) \geq g(\hat{\boldsymbol{\alpha}}_l) - 2\lambda_l \hat{\boldsymbol{\alpha}}_l' \mathbf{K} (\hat{\boldsymbol{\alpha}}_{l-1} - \hat{\boldsymbol{\alpha}}_l). \tag{24}$$

Likewise we have

$$g(\hat{\boldsymbol{\alpha}}_l) \geq g(\hat{\boldsymbol{\alpha}}_{l-1}) - 2\lambda_{l-1} \hat{\boldsymbol{\alpha}}_{l-1}' \mathbf{K} (\hat{\boldsymbol{\alpha}}_l - \hat{\boldsymbol{\alpha}}_{l-1}). \tag{25}$$

By summing up inequalities (24) and (25), we have

$$\lambda_{l-1} \hat{\boldsymbol{\alpha}}_{l-1}' \mathbf{K} (\hat{\boldsymbol{\alpha}}_l - \hat{\boldsymbol{\alpha}}_{l-1}) + \lambda_l \hat{\boldsymbol{\alpha}}_l' \mathbf{K} (\hat{\boldsymbol{\alpha}}_{l-1} - \hat{\boldsymbol{\alpha}}_l) \geq 0,$$

which is equivalent to

$$\left( \hat{\boldsymbol{\alpha}}_l - \frac{\lambda_{l-1} + \lambda_l}{2\lambda_l} \hat{\boldsymbol{\alpha}}_{l-1} \right)' \mathbf{K} \left( \hat{\boldsymbol{\alpha}}_l - \frac{\lambda_{l-1} + \lambda_l}{2\lambda_l} \hat{\boldsymbol{\alpha}}_{l-1} \right) \leq \frac{(\lambda_{l-1} - \lambda_l)^2}{4\lambda_l^2} \hat{\boldsymbol{\alpha}}_{l-1}' \mathbf{K} \hat{\boldsymbol{\alpha}}_{l-1}. \tag{26}$$

Thus inequality (26) serves as a bound for $\hat{\boldsymbol{\alpha}}_l$ when $\hat{\boldsymbol{\alpha}}_{l-1}$ is known. Let $\boldsymbol{\delta} = \hat{\boldsymbol{\alpha}}_l - \frac{\lambda_{l-1}+\lambda_l}{2\lambda_l}\hat{\boldsymbol{\alpha}}_{l-1}$. For each $i$, inequality (26) gives

$$
\begin{aligned}
y_i \mathbf{K}_i' \hat{\boldsymbol{\alpha}}_l &\leq \max_{\left\{\boldsymbol{\delta}:\boldsymbol{\delta}'\mathbf{K}\boldsymbol{\delta} \leq \frac{(\lambda_{l-1}-\lambda_l)^2}{4\lambda_l^2}\hat{\boldsymbol{\alpha}}_{l-1}'\mathbf{K}\hat{\boldsymbol{\alpha}}_{l-1}\right\}} y_i \mathbf{K}_i' \left(\frac{\lambda_{l-1}+\lambda_l}{2\lambda_l}\hat{\boldsymbol{\alpha}}_{l-1} + \boldsymbol{\delta}\right) \\
&\leq \frac{\lambda_{l-1}+\lambda_l}{2\lambda_l} y_i \mathbf{K}_i' \hat{\boldsymbol{\alpha}}_{l-1} + \max_{\left\{\boldsymbol{\delta}:\boldsymbol{\delta}'\mathbf{K}\boldsymbol{\delta} \leq \frac{(\lambda_{l-1}-\lambda_l)^2}{4\lambda_l^2}\hat{\boldsymbol{\alpha}}_{l-1}'\mathbf{K}\hat{\boldsymbol{\alpha}}_{l-1}\right\}} |\mathbf{K}_i'\boldsymbol{\delta}| \\
&= \frac{\lambda_{l-1}+\lambda_l}{2\lambda_l} y_i \mathbf{K}_i' \hat{\boldsymbol{\alpha}}_{l-1} + \max_{\left\{\boldsymbol{\delta}:\boldsymbol{\delta}'\mathbf{K}\boldsymbol{\delta} \leq \frac{(\lambda_{l-1}-\lambda_l)^2}{4\lambda_l^2}\hat{\boldsymbol{\alpha}}_{l-1}'\mathbf{K}\hat{\boldsymbol{\alpha}}_{l-1}\right\}} \left|\left\langle \mathbf{K}_i'\mathbf{K}^{-\frac{1}{2}}, \mathbf{K}^{\frac{1}{2}}\boldsymbol{\delta}\right\rangle\right| \\
&\leq \frac{\lambda_{l-1}+\lambda_l}{2\lambda_l} y_i \mathbf{K}_i' \hat{\boldsymbol{\alpha}}_{l-1} + \frac{\lambda_{l-1}-\lambda_l}{2\lambda_l}\sqrt{B}\sqrt{\hat{\boldsymbol{\alpha}}_{l-1}'\mathbf{K}\hat{\boldsymbol{\alpha}}_{l-1}},
\end{aligned}
$$

(27)

where the last inequality is due to Cauchy-Schwartz inequality. Similarly we can show that

$$
\begin{aligned}
y_i \mathbf{K}_i' \hat{\boldsymbol{\alpha}}_l &\geq \min_{\left\{\boldsymbol{\delta}:\boldsymbol{\delta}'\mathbf{K}\boldsymbol{\delta} \leq \frac{(\lambda_{l-1}-\lambda_l)^2}{4\lambda_l^2}\hat{\boldsymbol{\alpha}}_{l-1}'\mathbf{K}\hat{\boldsymbol{\alpha}}_{l-1}\right\}} y_i \mathbf{K}_i' \left(\frac{\lambda_{l-1}+\lambda_l}{2\lambda_l}\hat{\boldsymbol{\alpha}}_{l-1} + \boldsymbol{\delta}\right) \\
&\geq \frac{\lambda_{l-1}+\lambda_l}{2\lambda_l} y_i \mathbf{K}_i' \hat{\boldsymbol{\alpha}}_{l-1} - \max_{\left\{\boldsymbol{\delta}:\boldsymbol{\delta}'\mathbf{K}\boldsymbol{\delta} \leq \frac{(\lambda_{l-1}-\lambda_l)^2}{4\lambda_l^2}\hat{\boldsymbol{\alpha}}_{l-1}'\mathbf{K}\hat{\boldsymbol{\alpha}}_{l-1}\right\}} |\mathbf{K}_i'\boldsymbol{\delta}| \\
&\geq \frac{\lambda_{l-1}+\lambda_l}{2\lambda_l} y_i \mathbf{K}_i' \hat{\boldsymbol{\alpha}}_{l-1} - \frac{\lambda_{l-1}-\lambda_l}{2\lambda_l}\sqrt{B}\sqrt{\hat{\boldsymbol{\alpha}}_{l-1}'\mathbf{K}\hat{\boldsymbol{\alpha}}_{l-1}}, \ \forall i.
\end{aligned}
$$

(28)

By observing $B/(2n\lambda_l) > 0$ in the definition of $a_{i,l}^-$ and $a_{i,l}^+$, inequalities (27) and (28) give inequality (6) for $j = 0$, i.e.,

$$
a_{i,l}^- \leq y_i \mathbf{K}_i' \hat{\boldsymbol{\alpha}}_l \leq a_{i,l}^+.
$$

For $j = 1, 2, \ldots, n$, we define $g^{[j]}(\boldsymbol{\alpha}) = \frac{1}{n}\sum_{i=1}^n (1 - \tilde{y}_i^{[j]}\mathbf{K}_i'\boldsymbol{\alpha})_+$. By using the similar approach of getting inequality (24), we have

$$
\begin{aligned}
g^{[j]}(\hat{\boldsymbol{\alpha}}_l) &\geq g^{[j]}(\hat{\boldsymbol{\alpha}}_l^{[-j]}) - 2\lambda_l \hat{\boldsymbol{\alpha}}_l^{[-j]'}\mathbf{K}(\hat{\boldsymbol{\alpha}}_l - \hat{\boldsymbol{\alpha}}_l^{[-j]}), \\
g(\hat{\boldsymbol{\alpha}}_l^{[-j]}) &\geq g(\hat{\boldsymbol{\alpha}}_l) - 2\lambda_l \hat{\boldsymbol{\alpha}}_l'\mathbf{K}(\hat{\boldsymbol{\alpha}}_l^{[-j]} - \hat{\boldsymbol{\alpha}}_l).
\end{aligned}
$$

By adding the two inequalities above together, we see

$$
(\hat{\boldsymbol{\alpha}}_l^{[-j]} - \hat{\boldsymbol{\alpha}}_l)'\mathbf{K}(\hat{\boldsymbol{\alpha}}_l^{[-j]} - \hat{\boldsymbol{\alpha}}_l) \leq \frac{1}{2n\lambda_l}\left|\left(1 - y_j\mathbf{K}_j'\hat{\boldsymbol{\alpha}}_l^{[-j]}\right)_+ - \left(1 - y_j\mathbf{K}_j'\hat{\boldsymbol{\alpha}}_l\right)_+\right|.
$$

Let $\boldsymbol{\xi} = \hat{\boldsymbol{\alpha}}_l^{[-j]} - \hat{\boldsymbol{\alpha}}_l$. Due to the Lipschitz continuity of the hinge loss and Cauchy-Schwartz inequality, we further have

$$
\boldsymbol{\xi}'\mathbf{K}\boldsymbol{\xi} \leq \frac{1}{2n\lambda_l}\left|\mathbf{K}_j'\boldsymbol{\xi}\right| \leq \frac{1}{2n\lambda_l}\left|\left\langle \mathbf{K}_j'\mathbf{K}^{-\frac{1}{2}}, \mathbf{K}^{\frac{1}{2}}\boldsymbol{\xi}\right\rangle\right| \leq \frac{\sqrt{B}}{2n\lambda_l}\sqrt{\boldsymbol{\xi}'\mathbf{K}\boldsymbol{\xi}},
$$

which implies $\sqrt{\boldsymbol{\xi}'\mathbf{K}\boldsymbol{\xi}} \leq \sqrt{B}/(2n\lambda_l)$. For any $i \neq j$,

$$
\begin{aligned}
y_i \mathbf{K}_i' \hat{\boldsymbol{\alpha}}_l^{[-j]} &= y_i \mathbf{K}_i'(\hat{\boldsymbol{\alpha}}_l + \boldsymbol{\xi}) \\
&\leq y_i \mathbf{K}_i' \hat{\boldsymbol{\alpha}}_l + \max_{\boldsymbol{\xi}:\sqrt{\boldsymbol{\xi}'\mathbf{K}\boldsymbol{\xi}} \leq \sqrt{B}/(2n\lambda_l)} y_i \mathbf{K}_i'\boldsymbol{\xi} \\
&\leq y_i \mathbf{K}_i' \hat{\boldsymbol{\alpha}}_l + \frac{B}{2n\lambda_l} \\
&\leq \frac{\lambda_{l-1}+\lambda_l}{2\lambda_l} y_i \mathbf{K}_i' \hat{\boldsymbol{\alpha}}_{l-1} + \frac{\lambda_{l-1}-\lambda_l}{2\lambda_l}\sqrt{B}\sqrt{\hat{\boldsymbol{\alpha}}_{l-1}'\mathbf{K}\hat{\boldsymbol{\alpha}}_{l-1}} + \frac{B}{2n\lambda_l} \\
&= a_{i,l}^+,
\end{aligned}
$$

where the second to last inequality is from Cauchy-Schwartz inequality and the last inequality is due to inequality (27). We can similarly show $y_i \mathbf{K}'_i \hat{\boldsymbol{\alpha}}_l^{[-j]} \geq a_{i,l}^-$ for each $i \neq j$ and thus complete the proof of inequality (6).

For $i = j$, $\hat{\alpha}_{i,l}^{[-j]} = 0$. For $i \neq j$, by the definition of $\mathcal{L}$ and $\mathcal{R}$ we have $y_i \mathbf{K}'_i \hat{\boldsymbol{\alpha}}_l^{[-j]} < 1$ when $i \in \mathcal{L}$ and $y_i \mathbf{K}'_i \hat{\boldsymbol{\alpha}}_l^{[-j]} > 1$ when $i \in \mathcal{R}$, thus the proof is completed due to expression (5).

### A.3   Proof of Lemma 3.1

The proof of Lemma 3.1 is similar to the proof of Theorem 2.2. For each $j = 0, 1, \ldots, n$, the sub-gradient optimality condition of problem (13) with respect to $\beta_{0,l}^{[-j]}$ and each $\mathbf{K}'_i \boldsymbol{\alpha}_l^{[-j]}$ gives

$$
\begin{aligned}
& 0 \in \sum_{i=1}^n \tilde{y}_i^{[j]} \partial L \left( \tilde{y}_i^{[j]} (\hat{\beta}_{0,l}^{[-j]} + \mathbf{K}'_i \hat{\boldsymbol{\alpha}}_l^{[-j]}) \right), \\
& 0 \in \frac{1}{n} \tilde{y}_i^{[j]} \partial L \left( \tilde{y}_i^{[j]} (\hat{\beta}_{0,l}^{[-j]} + \mathbf{K}'_i \hat{\boldsymbol{\alpha}}_l^{[-j]}) \right) + 2\lambda_l \hat{\alpha}_{i,l}^{[-j]}, \ \forall i.
\end{aligned}
\tag{29}
$$

For any $\beta_0$ and $\boldsymbol{\alpha}$, define $\tilde{g}_j(\beta_0, \boldsymbol{\alpha}) = \frac{1}{n} \sum_{i=1}^n [1 - \tilde{y}_i^{[j]}(\beta_0 + \mathbf{K}'_i \boldsymbol{\alpha})]_+$. The convexity of $\tilde{g}_j$ implies

$$
\tilde{g}_j(\hat{\beta}_{0,l-1}^{[-j]}, \hat{\boldsymbol{\alpha}}_{l-1}^{[-j]}) \geq \tilde{g}_j(\hat{\beta}_{0,l}^{[-j]}, \hat{\boldsymbol{\alpha}}_l^{[-j]}) + \frac{1}{n} \sum_{i=1}^n \tilde{y}_i^{[j]} v_{ij} (\hat{\beta}_{0,l-1}^{[-j]} - \hat{\beta}_{0,l}^{[-j]}) + \frac{1}{n} \sum_{i=1}^n \tilde{y}_i^{[j]} v_{ij} \mathbf{K}'_i (\hat{\boldsymbol{\alpha}}_{l-1}^{[-j]} - \hat{\boldsymbol{\alpha}}_l^{[-j]}),
\tag{30}
$$

for any $v_{ij} \in \partial L(\tilde{y}_i^{[j]}(\hat{\beta}_{0,l}^{[-j]} + \mathbf{K}'_i \hat{\boldsymbol{\alpha}}_l^{[-j]}))$. From expressions (29), we let $v_{ij} = -2\lambda_l n \tilde{y}_i^{[j]} \hat{\alpha}_{i,l}^{[-j]}$ and then $\sum_{i=1}^n \tilde{y}_i^{[j]} v_{ij} = 0$ for each $j$. Subsequently inequality (30) implies

$$
\tilde{g}_j(\hat{\beta}_{0,l-1}^{[-j]}, \hat{\boldsymbol{\alpha}}_{l-1}^{[-j]}) \geq \tilde{g}_j(\hat{\beta}_{0,l}^{[-j]}, \hat{\boldsymbol{\alpha}}_l^{[-j]}) - 2\lambda_l \hat{\boldsymbol{\alpha}}_l^{[-j]'} \mathbf{K} (\hat{\boldsymbol{\alpha}}_{l-1}^{[-j]} - \hat{\boldsymbol{\alpha}}_l^{[-j]}).
$$

Similarly we have

$$
\tilde{g}_j(\hat{\beta}_{0,l}^{[-j]}, \hat{\boldsymbol{\alpha}}_l^{[-j]}) \geq \tilde{g}_j(\hat{\beta}_{0,l-1}^{[-j]}, \hat{\boldsymbol{\alpha}}_{l-1}^{[-j]}) - 2\lambda_{l-1} \hat{\boldsymbol{\alpha}}_{l-1}^{[-j]'} \mathbf{K} (\hat{\boldsymbol{\alpha}}_l^{[-j]} - \hat{\boldsymbol{\alpha}}_{l-1}^{[-j]}).
$$

By adding the above two inequalities, we have

$$
\left( \hat{\boldsymbol{\alpha}}_l^{[-j]} - \frac{\lambda_{l-1} + \lambda_l}{2\lambda_l} \hat{\boldsymbol{\alpha}}_{l-1}^{[-j]} \right)' \mathbf{K} \left( \hat{\boldsymbol{\alpha}}_l^{[-j]} - \frac{\lambda_{l-1} + \lambda_l}{2\lambda_l} \hat{\boldsymbol{\alpha}}_{l-1}^{[-j]} \right) \leq \frac{(\lambda_{l-1} - \lambda_l)^2}{4\lambda_l^2} \hat{\boldsymbol{\alpha}}_{l-1}^{[-j]'} \mathbf{K} \hat{\boldsymbol{\alpha}}_{l-1}^{[-j]}.
$$

Let $\boldsymbol{\delta} = \hat{\boldsymbol{\alpha}}_l^{[-j]} - \frac{\lambda_{l-1} + \lambda_l}{2\lambda_l} \hat{\boldsymbol{\alpha}}_{l-1}^{[-j]}$, and then we see for each $i \neq j$,

$$
\begin{aligned}
y_i \mathbf{K}'_i \hat{\boldsymbol{\alpha}}_l^{[-j]} & \leq \max_{\left\{ \boldsymbol{\delta} : \boldsymbol{\delta}' \mathbf{K} \boldsymbol{\delta} \leq \frac{(\lambda_{l-1} - \lambda_l)^2}{4\lambda_l^2} \hat{\boldsymbol{\alpha}}_{l-1}^{[-j]'} \mathbf{K} \hat{\boldsymbol{\alpha}}_{l-1}^{[-j]} \right\}} y_i \mathbf{K}'_i \left( \frac{\lambda_{l-1} + \lambda_l}{2\lambda_l} \hat{\boldsymbol{\alpha}}_{l-1}^{[-j]} + \boldsymbol{\delta} \right) \\
& \leq \frac{\lambda_{l-1} + \lambda_l}{2\lambda_l} y_i \mathbf{K}'_i \hat{\boldsymbol{\alpha}}_{l-1}^{[-j]} + \max_{\left\{ \boldsymbol{\delta} : \boldsymbol{\delta}' \mathbf{K} \boldsymbol{\delta} \leq \frac{(\lambda_{l-1} - \lambda_l)^2}{4\lambda_l^2} \hat{\boldsymbol{\alpha}}_{l-1}^{[-j]'} \mathbf{K} \hat{\boldsymbol{\alpha}}_{l-1}^{[-j]} \right\}} |\mathbf{K}'_i \boldsymbol{\delta}| \\
& = \frac{\lambda_{l-1} + \lambda_l}{2\lambda_l} y_i \mathbf{K}'_i \hat{\boldsymbol{\alpha}}_{l-1}^{[-j]} + \max_{\left\{ \boldsymbol{\delta} : \boldsymbol{\delta}' \mathbf{K} \boldsymbol{\delta} \leq \frac{(\lambda_{l-1} - \lambda_l)^2}{4\lambda_l^2} \hat{\boldsymbol{\alpha}}_{l-1}^{[-j]'} \mathbf{K} \hat{\boldsymbol{\alpha}}_{l-1}^{[-j]} \right\}} \left| \left\langle \mathbf{K}'_i \mathbf{K}^{-\frac{1}{2}}, \mathbf{K}^{\frac{1}{2}} \boldsymbol{\delta} \right\rangle \right| \\
& \leq \frac{\lambda_{l-1} + \lambda_l}{2\lambda_l} y_i \mathbf{K}'_i \hat{\boldsymbol{\alpha}}_{l-1}^{[-j]} + \frac{\lambda_{l-1} - \lambda_l}{2\lambda_l} \sqrt{B} \sqrt{\hat{\boldsymbol{\alpha}}_{l-1}^{[-j]'} \mathbf{K} \hat{\boldsymbol{\alpha}}_{l-1}^{[-j]}} \\
& \leq \max_{\substack{j=0,1,\ldots,n \\ j \neq i}} \left\{ \frac{\lambda_{l-1} + \lambda_l}{2\lambda_l} y_i \mathbf{K}'_i \hat{\boldsymbol{\alpha}}_{l-1}^{[-j]} + \frac{\lambda_{l-1} - \lambda_l}{2\lambda_l} \sqrt{B} \sqrt{\hat{\boldsymbol{\alpha}}_{l-1}^{[-j]'} \mathbf{K} \hat{\boldsymbol{\alpha}}_{l-1}^{[-j]}} \right\} \\
& = c_{i,l}^+.
\end{aligned}
$$

Likewise we can show $y_i \mathbf{K}'_i \hat{\boldsymbol{\alpha}}_l^{[-j]} \geq c_{i,l}^-$ and we thus prove inequality (15).

## A.4 Proof of Lemma 3.2

**Proof of (1).** Denote by $|\mathcal{S}|$ cardinality of a set $\mathcal{S}$. The definition of $\mathcal{S}_1(b)$ and $n_+(b)$ gives

$$
\begin{aligned}
\psi^+(b) &= \left( \sum_{i \in \mathcal{S}_1(b)} y_i \right) + n_+(b) + 1 \\
&= -|\{i : -b + c_{i,l}^+ < 1, \ y_i = -1\}| + |\{i : b + c_{i,l}^+ < 1, \ y_i = 1\}| \\
&\quad + |\{i : b + c_{i,l}^+ \geq 1, b + c_{i,l}^- \leq 1, \ y_i = 1\}| + 1 \\
&= -|\{i : -b + c_{i,l}^+ < 1, \ y_i = -1\}| + |\{i : b + c_{i,l}^+ < 1, \ y_i = 1\}| \\
&\quad - |\{i : b + c_{i,l}^+ < 1, \ y_i = 1\}| + |\{i : b + c_{i,l}^- \leq 1, \ y_i = 1\}| + 1 \\
&= -|\{i : -b + c_{i,l}^+ < 1, \ y_i = -1\}| + |\{i : b + c_{i,l}^- \leq 1, \ y_i = 1\}| + 1,
\end{aligned}
$$

which is non-increasing in $b$. We also find $\psi^-(b)$ non-increasing because

$$
\begin{aligned}
\psi^-(b) &= \left( \sum_{i \in \mathcal{S}_1(b)} y_i \right) - n_-(b) - 1 \\
&= -|\{i : -b + c_{i,l}^+ < 1, \ y_i = -1\}| + |\{i : b + c_{i,l}^+ < 1, \ y_i = 1\}| \\
&\quad - |\{i : -b + c_{i,l}^+ \geq 1, -b + c_{i,l}^- \leq 1, \ y_i = -1\}| - 1 \\
&= -|\{i : -b + c_{i,l}^+ < 1, \ y_i = -1\}| + |\{i : b + c_{i,l}^+ < 1, \ y_i = 1\}| \\
&\quad + |\{i : -b + c_{i,l}^+ < 1, \ y_i = -1\}| - |\{i : -b + c_{i,l}^- \leq 1, \ y_i = -1\}| - 1 \\
&= -|\{i : -b + c_{i,l}^- \leq 1, \ y_i = -1\}| + |\{i : b + c_{i,l}^+ < 1, \ y_i = 1\}| - 1.
\end{aligned}
$$

**Proof of (2).** For any $j = 0, 1, \ldots, n$, from inequality (15) and the definition of $\mathcal{S}_1(b)$ and $\mathcal{S}_2(b)$, we have $\partial L(y_i(b + \mathbf{K}_i' \hat{\boldsymbol{\alpha}}_l^{[-j]})) = -1$ if $i \in \mathcal{S}_1(b)$, and $\partial L(y_i(b + \mathbf{K}_i' \hat{\boldsymbol{\alpha}}_l^{[-j]})) = 0$ if $i \in \mathcal{S}_2(b)$. Also by the definition of each $\tilde{y}_i^{[j]}$, we see

$$
\tilde{y}_i^{[j]} \partial L(\tilde{y}_i^{[j]}(b + \mathbf{K}_i' \hat{\boldsymbol{\alpha}}_l^{[-j]})) = \begin{cases} -\tilde{y}_i^{[j]} & \text{if } i \in \mathcal{S}_1(b), \\ 0 & \text{if } i \in \mathcal{S}_2(b). \end{cases}
$$

Hence

$$
\begin{aligned}
&\sum_{i=1}^n \tilde{y}_i^{[j]} \partial L \left( \tilde{y}_i^{[j]}(b + \mathbf{K}_i' \hat{\boldsymbol{\alpha}}_l^{[-j]}) \right) \\
&= \sum_{i \in \mathcal{S}_1(b)} \tilde{y}_i^{[j]} \partial L \left( \tilde{y}_i^{[j]}(b + \mathbf{K}_i' \hat{\boldsymbol{\alpha}}_l^{[-j]}) \right) + \sum_{i \in \mathcal{S}_2(b)} \tilde{y}_i^{[j]} \partial L \left( \tilde{y}_i^{[j]}(b + \mathbf{K}_i' \hat{\boldsymbol{\alpha}}_l^{[-j]}) \right) \\
&\quad + \sum_{i \in (\mathcal{S}_1(b) \cup \mathcal{S}_2(b))^C} \tilde{y}_i^{[j]} \partial L \left( \tilde{y}_i^{[j]}(b + \mathbf{K}_i' \hat{\boldsymbol{\alpha}}_l^{[-j]}) \right) \\
&= \sum_{i \in \mathcal{S}_1(b)} (-\tilde{y}_i^{[j]}) + \sum_{i \in (\mathcal{S}_1(b) \cup \mathcal{S}_2(b))^C} \tilde{y}_i^{[j]} \partial L \left( \tilde{y}_i^{[j]}(b + \mathbf{K}_i' \hat{\boldsymbol{\alpha}}_l^{[-j]}) \right).
\end{aligned}
$$

When $i \in (\mathcal{S}_1(b) \cup \mathcal{S}_2(b))^C$, $\partial L(\tilde{y}_i^{[j]}(b + \mathbf{K}_i' \hat{\boldsymbol{\alpha}}_l^{[-j]})) \in [-1, 0]$, so

$$
\sum_{i \in (\mathcal{S}_1(b) \cup \mathcal{S}_2(b))^C} \tilde{y}_i^{[j]} \partial L \left( \tilde{y}_i^{[j]}(b + \mathbf{K}_i' \hat{\boldsymbol{\alpha}}_l^{[-j]}) \right) \in [-n_+(b), n_-(b)],
$$

which says

$$
\sum_{i \in \mathcal{S}_1(b)} \tilde{y}_i^{[j]} \in [-n_+(b), n_-(b)]
$$

is a necessary condition for $b = \hat{\beta}_{0,l}^{[-j]}$; otherwise, $0 \notin \sum_{i=1}^n \tilde{y}_i^{[j]} \partial L \left( \tilde{y}_i^{[j]}(b + \mathbf{K}_i' \hat{\boldsymbol{\alpha}}_l^{[-j]}) \right)$ and the sub-gradient optimality condition is violated.

From the definition of $\tilde{y}_i^{[j]}$, we see

$$\sum_{i \in \mathcal{S}_1(b)} y_i \in [-n_+(b) - 1, n_-(b) + 1] \tag{31}$$

is a necessary condition for $b = \hat{\beta}_{0,l}^{[-j]}$ for any $j$. This says that the violation of condition (31) implies that $b \neq \hat{\beta}_{0,l}^{[-j]}$ for any $j = 0, 1, \ldots, n$.

Therefore, if $\psi^+(b) = (\sum_{i \in \mathcal{S}_1(b)} y_i) + n_+(b) + 1 < 0$, then for any $b' > b$, $\psi^+(b') < 0$, that is, $\sum_{i \in \mathcal{S}_1(b')} y_i < -n_+(b') - 1$. This says $b > \hat{\beta}_{0,l}^{[-j]}$ for any $j$ by condition (31).

**Proof of (3).** If $\psi^-(b) = (\sum_{i \in \mathcal{S}_1(b)} y_i) - n_-(b) - 1 > 0$, then for any $b' < b$, $\psi^-(b') > 0$, that is, $\sum_{i \in \mathcal{S}_1(b')} y_i > n_-(b') + 1$. Condition (31) shows that $b < \hat{\beta}_{0,l}^{[-j]}$ for any $j = 0, 1, \ldots, n$.

### A.5 Proof of Lemma 3.3

The bi-section algorithm is detailed in Algorithm 2 in Section B. We first show $\psi^+(B^+) < 0$ and $\psi^-(B^-) > 0$. By the definition of $B^+$, we have $B^+ + c_{i,l}^- > 1$ for all $i$ such that $y_i = 1$, and $-B^+ + c_{i,l}^+ < 1$ for all $i$ such that $y_i = -1$. Thus we have $n_+(B^+) = 0$, $|\{i : B^+ + c_{i,l}^- \leq 1, \ y_i = 1\}| = 0$, and then $\psi^+(B^+) = -|\{i : -B^+ + c_{i,l}^+ < 1, \ y_i = -1\}| + 1 < 0$. Likewise, the definition of $B^-$ gives $|\{i : B^- + c_{i,l}^+ < 1, \ y_i = 1\}| - 1 > 0$ and $|\{i : -B^- + c_{i,l}^- \leq 1, \ y_i = -1\}| = 0$, and thus the definition of $\psi^-$ implies $\psi^-(B^-) > 0$.

In Algorithm 2, $a^+$ is initialized to be $B^+$ and $\psi^+(a^+) < 0$ always holds when $a^+$ is updated by some $b^+$ such that $\psi^+(b^+) < 0$. As $\beta_{0,l}^+$ is set to be $a^+$ when the algorithm converges, $\psi^+(\beta_{0,l}^+) < 0$, which shows $\beta_{0,l}^+ > \hat{\beta}_{0,l}^{[-j]}$ for any $j$ by (2) of Lemma 3.2.

Likewise, $c^-$ is initialized to be $B^-$ and $\psi^-(c^-) > 0$ always holds when $c^-$ is updated by some $b^-$ such that $\psi^+(b^-) > 0$. As $\beta_{0,l}^-$ is set to be $c^-$ when the algorithm converges, $\psi^-(\beta_{0,l}^-) > 0$, which shows $\beta_{0,l}^- < \hat{\beta}_{0,l}^{[-j]}$ for any $j$ by (3) of Lemma 3.2.

### A.6 Proof of Lemma 3.4

We first show inequality (18) for $j = 0$, which is equivalent to

$$\tilde{c}_{i,l}^- \leq y_i \mathbf{K}_i' \hat{\boldsymbol{\alpha}}_l \leq \tilde{c}_{i,l}^+.$$

Denote by $g(\beta_0, \boldsymbol{\alpha}) = \frac{1}{n} \sum_{i=1}^n (1 - y_i(\beta_0 + \mathbf{K}_i' \boldsymbol{\alpha}))_+$. The sub-gradient optimality condition of problem (12) with respect to $\beta_{0,l}$ and $\mathbf{K}\boldsymbol{\alpha}_l$ gives

$$0 \in \frac{1}{n} y_i \partial L(y_i(\hat{\beta}_{0,l} + \mathbf{K}_i' \hat{\boldsymbol{\alpha}}_l)) + 2\lambda_l \hat{\alpha}_{i,l}, \ \forall i,$$

$$0 \in \sum_{i=1}^n y_i \partial L(y_i(\hat{\beta}_{0,l} + \mathbf{K}_i' \hat{\boldsymbol{\alpha}}_l)). \tag{32}$$

The convexity of $g$ implies

$$g(\hat{\beta}_{0,l-1}, \hat{\boldsymbol{\alpha}}_{l-1}) \geq g(\hat{\beta}_{0,l}, \hat{\boldsymbol{\alpha}}_l) + \frac{1}{n} \sum_{i=1}^n y_i v_i (\hat{\beta}_{0,l-1} - \hat{\beta}_{0,l}) + \frac{1}{n} \sum_{i=1}^n y_i v_i \mathbf{K}_i' (\hat{\boldsymbol{\alpha}}_{l-1} - \hat{\boldsymbol{\alpha}}_l), \tag{33}$$

for any $v_i \in \partial L(y_i(\hat{\beta}_{0,l} + \mathbf{K}_i' \hat{\boldsymbol{\alpha}}_l))$. By expressions (32) and (33), setting $v_i = -2\lambda_l n y_i \hat{\alpha}_{i,l}$, we have

$$g(\hat{\beta}_{0,l-1}, \hat{\boldsymbol{\alpha}}_{l-1}) \geq g(\hat{\beta}_{0,l}, \hat{\boldsymbol{\alpha}}_l) - 2\lambda_l \hat{\boldsymbol{\alpha}}_l' \mathbf{K}(\hat{\boldsymbol{\alpha}}_{l-1} - \hat{\boldsymbol{\alpha}}_l).$$

We then use the same approach of getting inequality (27) in the proof of Theorem 2.2 to give

$$y_i \mathbf{K}_i' \hat{\boldsymbol{\alpha}}_l \leq \frac{\lambda_{l-1} + \lambda_l}{2\lambda_l} y_i \mathbf{K}_i' \hat{\boldsymbol{\alpha}}_{l-1} + \frac{\lambda_{l-1} - \lambda_l}{2\lambda_l} \sqrt{B} \sqrt{\hat{\boldsymbol{\alpha}}_{l-1}' \mathbf{K} \hat{\boldsymbol{\alpha}}_{l-1}}, \ \forall i,$$

$$y_i \mathbf{K}_i' \hat{\boldsymbol{\alpha}}_l \geq \frac{\lambda_{l-1} + \lambda_l}{2\lambda_l} y_i \mathbf{K}_i' \hat{\boldsymbol{\alpha}}_{l-1} - \frac{\lambda_{l-1} - \lambda_l}{2\lambda_l} \sqrt{B} \sqrt{\hat{\boldsymbol{\alpha}}_{l-1}' \mathbf{K} \hat{\boldsymbol{\alpha}}_{l-1}}, \ \forall i. \tag{34}$$

Thus inequality (18) is proved for $j = 0$.

We then define $g^{[j]}(\beta_0, \boldsymbol{\alpha}) = \frac{1}{n}\sum_{i=1}^{n}(1 - \tilde{y}_i^{[j]}(\beta_0 + \mathbf{K}_i'\boldsymbol{\alpha}))_+$ for each $j$. By using the same approach of getting inequality (24), we have

$$g^{[j]}(\hat{\beta}_{0,l}, \hat{\boldsymbol{\alpha}}_l) \geq g^{[j]}(\hat{\beta}_{0,l}^{[-j]}, \hat{\boldsymbol{\alpha}}_l^{[-j]}) - 2\lambda_l \hat{\boldsymbol{\alpha}}_l^{[-j]'}\mathbf{K}(\hat{\boldsymbol{\alpha}}_l - \hat{\boldsymbol{\alpha}}_l^{[-j]}),$$

$$g(\hat{\beta}_{0,l}^{[-j]}, \hat{\boldsymbol{\alpha}}_l^{[-j]}) \geq g(\hat{\beta}_{0,l}, \hat{\boldsymbol{\alpha}}_l) - 2\lambda_l \hat{\boldsymbol{\alpha}}_l'\mathbf{K}(\hat{\boldsymbol{\alpha}}_l^{[-j]} - \hat{\boldsymbol{\alpha}}_l).$$

By adding the two inequalities above together, we obtain

$$(\hat{\boldsymbol{\alpha}}_l^{[-j]} - \hat{\boldsymbol{\alpha}}_l)'\mathbf{K}(\hat{\boldsymbol{\alpha}}_l^{[-j]} - \hat{\boldsymbol{\alpha}}_l) \leq \frac{1}{2n\lambda_l}\left|\left(1 - y_j\hat{\beta}_{0,l}^{[-j]} - y_j\mathbf{K}_j'\hat{\boldsymbol{\alpha}}_l^{[-j]}\right)_+ - \left(1 - y_j\hat{\beta}_{0,l} - y_j\mathbf{K}_j'\hat{\boldsymbol{\alpha}}_l\right)_+\right|. \tag{35}$$

Let $\boldsymbol{\vartheta} = \hat{\boldsymbol{\alpha}}_l^{[-j]} - \hat{\boldsymbol{\alpha}}_l$. Due to the Lipschitz continuity of the hinge loss and inequality (17), we see

$$\begin{aligned}
&\left|\left(1 - y_j\hat{\beta}_{0,l}^{[-j]} - y_j\mathbf{K}_j'\hat{\boldsymbol{\alpha}}_l^{[-j]}\right)_+ - \left(1 - y_j\hat{\beta}_{0,l} - y_j\mathbf{K}_j'\hat{\boldsymbol{\alpha}}_l\right)_+\right| \\
&\leq \left|\left(y_j\hat{\beta}_{0,l}^{[-j]} + y_j\mathbf{K}_j'\hat{\boldsymbol{\alpha}}_l^{[-j]}\right) - \left(y_j\hat{\beta}_{0,l} + y_j\mathbf{K}_j'\hat{\boldsymbol{\alpha}}_l\right)\right| \\
&\leq |\hat{\beta}_{0,l}^{[-j]} - \hat{\beta}_{0,l}| + \left|\mathbf{K}_j'\boldsymbol{\vartheta}\right| \\
&\leq \beta_{0,l}^+ - \beta_{0,l}^- + \left|\mathbf{K}_j'\boldsymbol{\vartheta}\right| \\
&\leq \beta_{0,l}^+ - \beta_{0,l}^- + \left|\left\langle\mathbf{K}_j'\mathbf{K}^{-\frac{1}{2}}, \mathbf{K}^{\frac{1}{2}}\boldsymbol{\vartheta}\right\rangle\right| \\
&\leq \beta_{0,l}^+ - \beta_{0,l}^- + \sqrt{B}\sqrt{\boldsymbol{\vartheta}'\mathbf{K}\boldsymbol{\vartheta}},
\end{aligned} \tag{36}$$

where the last inequality is from Cauchy-Schwartz inequality. Let $c_l = \beta_{0,l}^+ - \beta_{0,l}^-$. By inequalities (35) and (36), we see,

$$\boldsymbol{\vartheta}'\mathbf{K}\boldsymbol{\vartheta} \leq \frac{1}{2n\lambda_l}(c_l + \sqrt{B}\sqrt{\boldsymbol{\vartheta}'\mathbf{K}\boldsymbol{\vartheta}}),$$

which gives $\boldsymbol{\vartheta} \in \mathcal{W}$ where

$$\mathcal{W} = \left\{\boldsymbol{\vartheta} : \sqrt{\boldsymbol{\vartheta}'\mathbf{K}\boldsymbol{\vartheta}} \leq \sqrt{\frac{B}{16n^2\lambda_l^2} + \frac{c_l}{2n\lambda_l}} + \frac{\sqrt{B}}{4n\lambda_l}\right\}.$$

It follows that

$$\max_{\boldsymbol{\vartheta}\in\mathcal{W}} |y_i\mathbf{K}_i'\boldsymbol{\vartheta}| \leq \max_{\boldsymbol{\vartheta}\in\mathcal{W}}\left|\left\langle\mathbf{K}_j'\mathbf{K}^{-\frac{1}{2}}, \mathbf{K}^{\frac{1}{2}}\boldsymbol{\vartheta}\right\rangle\right| \leq \max_{\boldsymbol{\vartheta}\in\mathcal{W}}\sqrt{B}\sqrt{\boldsymbol{\vartheta}'\mathbf{K}\boldsymbol{\vartheta}} \leq \sqrt{\frac{B^2}{16n^2\lambda_l^2} + \frac{c_lB}{2n\lambda_l}} + \frac{B}{4n\lambda_l}. \tag{37}$$

For any $j \neq i$, from inequalities (34) and (37) we see

$$y_i\mathbf{K}_i'\hat{\boldsymbol{\alpha}}_l^{[-j]} = y_i\mathbf{K}_i'(\hat{\boldsymbol{\alpha}}_l + \boldsymbol{\vartheta}) \leq y_i\mathbf{K}_i'\hat{\boldsymbol{\alpha}}_l + \max_{\boldsymbol{\vartheta}\in\mathcal{W}} |y_i\mathbf{K}_i'\boldsymbol{\vartheta}| \leq \tilde{c}_{i,l}^+,$$

$$y_i\mathbf{K}_i'\hat{\boldsymbol{\alpha}}_l^{[-j]} = y_i\mathbf{K}_i'(\hat{\boldsymbol{\alpha}}_l + \boldsymbol{\vartheta}) \geq y_i\mathbf{K}_i'\hat{\boldsymbol{\alpha}}_l - \max_{\boldsymbol{\vartheta}\in\mathcal{W}} |y_i\mathbf{K}_i'\boldsymbol{\vartheta}| \geq \tilde{c}_{i,l}^-.$$

## A.7    Proof of Theorem 3.5

The sub-gradient optimality condition of problem (13) with respect to $\beta_{0,l}$ and $\mathbf{K}\boldsymbol{\alpha}_l$ gives

$$0 \in \frac{1}{n}\tilde{y}_i^{[j]}\partial L\left(\tilde{y}_i^{[j]}(\hat{\beta}_{0,l}^{[-j]} + \mathbf{K}_i'\hat{\boldsymbol{\alpha}}_l^{[-j]})\right) + 2\lambda_l\hat{\alpha}_{i,l}^{[-j]}, \ \forall i. \tag{38}$$

For any $j = 0, 1, \ldots, n$, we see $\hat{\alpha}_{i,l}^{[-i]} = 0$ if $i = j$. Thus we focus on $i \neq j$, where $y_i = \tilde{y}_i^{[j]}$ by its definition. If $i \in \tilde{\mathcal{L}}$, then inequality (20) implies $\tilde{y}_i^{[j]}\hat{\beta}_{0,l}^{[-j]} + \hat{c}_{i,l}^+ < 1$, then by expression (38), $\partial L(\tilde{y}_i^{[j]}(\hat{\beta}_{0,l}^{[-j]} + \mathbf{K}_i'\hat{\boldsymbol{\alpha}}_l^{[-j]})) = -1$ and $\hat{\alpha}_{i,l}^{[-j]} = \tilde{y}_i^{[j]}/(2n\lambda_l)$.

If $i \in \tilde{\mathcal{R}}$, then inequality (20) implies $\tilde{y}_i^{[j]}\hat{\beta}_{0,l}^{[-j]} + \hat{c}_{i,l}^- > 1$, then expression (38) gives $\partial L(\tilde{y}_i^{[j]}(\hat{\beta}_{0,l}^{[-j]} + \mathbf{K}_i'\hat{\boldsymbol{\alpha}}_l^{[-j]})) = 0$ and $\hat{\alpha}_{i,l}^{[-j]} = 0$.

# B   Pseudocode

In Algorithm 3, we summarize the consolidated CV algorithm for solving the general SVM problems with the bias term introduced in Section 3.

---

**Algorithm 3** Consolidated cross-validation for general SVM problems

---

**Input**: $\lambda_1 > \lambda_2 > ... > \lambda_L, \mathbf{K}, \mathbf{y}$.

1:  Obtain
$$(\hat{\beta}_{01}, \hat{\boldsymbol{\alpha}}_1) = \underset{\beta_0 \in I\!R, \boldsymbol{\alpha} \in I\!R^n}{\operatorname{argmin}} \frac{1}{n} \sum_{i=1}^n (1 - \beta_0 - y_i \mathbf{K}_i' \boldsymbol{\alpha})_+ + \lambda_1 \boldsymbol{\alpha}' \mathbf{K} \boldsymbol{\alpha}.$$

2: **for** $l = 2, 3, \ldots, L$ **do**

3:     Obtain $c_{i,l}^-$ and $c_{i,l}^+$ from equations (14) for each $i$.

4:     Call Algorithm 2 with $c_{i,l}^-$ and $c_{i,l}^+$ to obtain $\beta_{0,l}^+$ and $\beta_{0,l}^-$.

5:     Obtain $\tilde{c}_{i,l}^-$ and $\tilde{c}_{i,l}^+$ from Lemma 3.4 for each $i$.

6:     Obtain $\hat{c}_{i,l}^-$ and $\hat{c}_{i,l}^+$ from equation (19) for each $i$.

7:     Call Algorithm 2 with $\hat{c}_{i,l}^-$ and $\hat{c}_{i,l}^+$ to obtain $\tilde{\beta}_{0,l}^+$ and $\tilde{\beta}_{0,l}^-$.

8:     Construct the sets $\tilde{\mathcal{L}}$ and $\tilde{\mathcal{R}}$ according to Theorem 3.5. Let $\mathcal{S} = (\tilde{\mathcal{L}} \cup \tilde{\mathcal{R}})^C$.

9:     Construct the matrices $\boldsymbol{\Gamma}$ and $\boldsymbol{\Sigma}$.

10:    **for** $j = 0, 1, \ldots, n$ **do**

11:      **if** $j > 0$ and $\hat{\alpha}_{j,l} = 0$ **then**

12:        Obtain $(\hat{\beta}_{0l}^{[-j]}, \hat{\boldsymbol{\alpha}}_l^{[-j]}) = (\hat{\beta}_{0l}, \hat{\boldsymbol{\alpha}}_l)$.

13:      **else**

14:        Construct the vector $\bar{\mathbf{y}}^{[j]}$.

15:        Obtain $\hat{\beta}_{0l}^{[-j]}$ and $\hat{\boldsymbol{\eta}}_l^{[-j]}$ by solving

$$\min_{\beta_0 \in I\!R, \boldsymbol{\eta} \in I\!R^{n_s}} \left[ \frac{1}{n} \sum_{i=1}^n \left( 1 - \tilde{y}_i^{[j]} (\beta_{0l}^{[-j]} + \boldsymbol{\Gamma}_i' \boldsymbol{\eta} + \frac{1}{2n\lambda_l} \mathbf{K}_i' \bar{\mathbf{y}}^{[j]}) \right)_+ + \frac{1}{n} \bar{\mathbf{y}}^{[j]'} \boldsymbol{\Gamma} \boldsymbol{\eta} + \lambda_l \boldsymbol{\eta}' \boldsymbol{\Sigma} \boldsymbol{\eta} \right].$$

16:        Obtain $\hat{\boldsymbol{\alpha}}_l^{[-j]}$ from expression (7) with $\tilde{\mathcal{L}}$ and $\tilde{\mathcal{R}}$.

17:      **end if**

18:    **end for**

19: **end for**

**Output**: $\hat{\beta}_{0l}, \hat{\boldsymbol{\alpha}}_l, \hat{\beta}_{0l}^{[-j]}$, and $\hat{\boldsymbol{\alpha}}_l^{[-j]}$, for each $j = 1, 2, \ldots, n$ and $l = 1, 2, \ldots, L$.

---

# C   Scaling Consolidated CV to Large-Scale Data Analysis

Although the kernel SVM is one of the most powerful nonlinear learning algorithms with diverse applications, one of its computational challenges is that storage and computation of the kernel matrix can be very expensive. To further improve scalability, we can incorporate kernel approximation into the existing consolidated CV algorithm. Specifically, random features (Rahimi and Recht, 2007) or Nyström subsampling (Rudi et al., 2015) can be applied in the exact leave-one-out formula of the SVM to find a low-cost approximation of the kernel matrix. Integrating these approximation techniques into our methods can further improve the numerical performance of `ccvsvm`.

## C.1   Consolidated CV with Nyström approaches

In this section, we describe how to incorporate Nyström approaches into `ccvsvm`. Let $\hat{f}(\mathbf{x})$ be the prediction function fitted on the training data. Let $\hat{f}^{[-j]}(\mathbf{x})$ be the prediction function fitted on the training data with the $j$th sample removed in the LOOCV procedure. For sake of presentation, we define $\hat{f}^{[-0]}(\mathbf{x}) = \hat{f}(\mathbf{x})$ to unify the notation of the training and the tuning of the SVM.

We have that $\hat{f}^{[-j]}(\mathbf{x}) = \sum_{i \neq j} \tilde{\alpha}_{i,l}^{[-j]} K(\mathbf{x}_i, \mathbf{x})$, where $\tilde{\boldsymbol{\alpha}}_l^{[-j]} = (\tilde{\alpha}_{1,l}^{[-j]}, \ldots, \tilde{\alpha}_{n,l}^{[-j]})'$ corresponds to the solution of (3). According to Lemma 2.1, we know that alternatively $\hat{f}^{[-j]}(\mathbf{x})$ can be obtained by $\hat{f}^{[-j]}(\mathbf{x}) = \sum_{i=1}^n \hat{\alpha}_{i,l}^{[-j]} K(\mathbf{x}_i, \mathbf{x})$, where $\hat{\boldsymbol{\alpha}}_l^{[-j]} = (\hat{\alpha}_{1,l}^{[-j]}, \ldots, \hat{\alpha}_{n,l}^{[-j]})'$ is obtained by solving a

surrogate problem (4) with the full dataset. This is due to the result of Lemma 2.1 that $\hat{\boldsymbol{\alpha}}_l^{[-j]} = (\tilde{\alpha}_{1,l}^{[-j]}, \ldots, \tilde{\alpha}_{j-1,l}^{[-j]}, 0, \tilde{\alpha}_{j,l}^{[-j]}, \ldots, \tilde{\alpha}_{n-1,l}^{[-j]})'$.

We can perform Nyström approximation of $\hat{f}^{[-j]}(\mathbf{x})$ to further improve the numerical performance. Specifically, we have $\{\mathbf{x}_1, \ldots, \mathbf{x}_n\}$ as $n$ observations of the training set. Let $\{\tilde{\mathbf{x}}_1, \ldots, \tilde{\mathbf{x}}_m\}$ be a subset of $m$ randomly selected observations ($m \leq n$) from the training set. Define an $n \times m$ matrix $\mathbf{K}_{nm}$ with $(\mathbf{K}_{nm})_{ij} = K(\mathbf{x}_i, \tilde{\mathbf{x}}_j)$ and let $\mathbf{K}_{mm}$ be an $m \times m$ matrix with $(\mathbf{K}_{mm})_{jk} = K(\tilde{\mathbf{x}}_j, \tilde{\mathbf{x}}_k)$ for $i \in \{1, \ldots, n\}$ and $j, k \in \{1, \ldots, m\}$. We can apply Nyström approximation $\hat{f}^{[-j]}(\mathbf{x}) \approx \sum_{i=1}^m \hat{\beta}_{i,l}^{[-j]} K(\tilde{\mathbf{x}}_i, \mathbf{x})$ where $\hat{\boldsymbol{\beta}}_l^{[-j]} = (\hat{\beta}_{1,l}^{[-j]}, \ldots, \hat{\beta}_{m,l}^{[-j]})'$ is the solution of the minimization problem:

$$\hat{\boldsymbol{\beta}}_l^{[-j]} = \operatorname*{argmin}_{\boldsymbol{\beta} \in \mathbb{R}^m} \left[ \frac{1}{n} \sum_{i=1}^n \left( 1 - \tilde{y}_i^{[j]} (\mathbf{K}_{nm})_i' \boldsymbol{\beta} \right)_+ + \lambda_l \boldsymbol{\beta}' \mathbf{K}^{mm} \boldsymbol{\beta} \right], \tag{39}$$

where $(\mathbf{K}_{nm})_i$ is the $i$th row of $\mathbf{K}_{nm}$. Compared with (4) which involves the full kernel matrix $\mathbf{K}$, (39) involves smaller matrix $\mathbf{K}_{nm}$ and $\mathbf{K}_{mm}$. With the introduction of $\boldsymbol{\gamma} = (\mathbf{K}_{mm})^{1/2} \boldsymbol{\beta}$ and $\mathbf{z}_i = (\mathbf{K}_{nm})_i' (\mathbf{K}_{mm}^+)^{1/2}$, where $\mathbf{K}_{mm}^+$ is the Moore–Penrose inverse of matrix $\mathbf{K}_{mm}$, problem (39) can be further convert into a ridge penalized linear problem with the hinge loss:

$$\hat{\boldsymbol{\gamma}}_l^{[-j]} = \operatorname*{argmin}_{\boldsymbol{\gamma} \in \mathbb{R}^m} \left[ \frac{1}{n} \sum_{i=1}^n \left( 1 - \tilde{y}_i^{[j]} \mathbf{z}_i \boldsymbol{\gamma} \right)_+ + \lambda_l \|\boldsymbol{\gamma}\|_2^2 \right]. \tag{40}$$

As a remark, the above Nyström approach is achieved in a consolidated way for the complete data solution $\hat{f}$ and all LOOCV solutions $\hat{f}^{[-j]}$, because the kernel matrix is the same due to the exact leave-one-out formula.

## C.2 Consolidated CV with random features

Alternatively, one can use random features (Rahimi and Recht, 2007) to approximate the kernel matrix. Suppose that we consider shift-invariant kernels that satisfy $K(\mathbf{x}, \mathbf{y}) = K(\mathbf{x} - \mathbf{y})$. In this work we use the radial kernel $K(\mathbf{x}, \mathbf{y}) = \exp(-\sigma \|\mathbf{x} - \mathbf{y}\|_2^2)$. The kernel can be approximated by $K(\mathbf{x}, \mathbf{y}) \approx \langle \varphi(\mathbf{x}), \varphi(\mathbf{y}) \rangle$, where an explicit randomized feature mapping $\varphi : \mathbb{R}^p \to \mathbb{R}^m$ is obtained by sampling from a distribution defined by the inverse Fourier transformation. Specifically, $\varphi(\mathbf{x}) = \cos(\omega' \mathbf{x} + b)$ where $\omega$ is drawn from $\mathrm{N}(0, 2\sigma)$ and $b$ is drawn uniformly from $[0, 2\pi]$. In order to to achieve computational efficiency, the number of random features $m$ is chosen to be larger than the original sample dimension $p$ but much smaller than the sample size $n$. We can use random features to approximate the leave-one-out prediction function $\hat{f}^{[-j]}(\mathbf{x}) \approx (\hat{\boldsymbol{\gamma}}_l^{[-j]})' \varphi(\mathbf{x})$. Here the coefficient $\hat{\boldsymbol{\gamma}}_l^{[-j]}$ can be obtained by solving the following approximate version of problem

$$\hat{\boldsymbol{\gamma}}_l^{[-j]} = \operatorname*{argmin}_{\boldsymbol{\gamma} \in \mathbb{R}^m} \left[ \frac{1}{n} \sum_{i=1}^n \left( 1 - \tilde{y}_i^{[j]} \mathbf{z}_i \boldsymbol{\gamma} \right)_+ + \lambda_l \|\boldsymbol{\gamma}\|_2^2 \right], \tag{41}$$

where $\mathbf{z}_i = \varphi(\mathbf{x}_i)'$ is the random features for the $i$th sample. We can see that (40) from the Nyström approach and (41) from the random-feature approach essentially share the same form, except that $\mathbf{z}_i$'s in the two problems represent different variables.

## C.3 Consolidated algorithm for solving problem (40) and (41)

In the previous sections, we have shown that both Nyström approximation and random features transform the original kernel SVM into linear SVM problems, i.e., (40) and (41). We now give a consolidated algorithm to solve the problem for all $j = 0, 1, 2, \ldots, n$.

With a given small $\tau$, we first give the smoothed SVM loss,

$$L_\tau(u) = \begin{cases} 0 & u \geq 1 + \tau, \\ (u - (1 + \tau))^2/(4\tau) & 1 - \tau < u < 1 + \tau, \\ 1 - u & u \leq 1 - \tau. \end{cases}$$

For each $j = 0, 1, 2, \ldots, n$, we develop a proximal gradient descent algorithm which updates $\boldsymbol{\gamma}^{(-j,t+1)}$ by

$$\boldsymbol{\gamma}^{(-j,t+1)} = \boldsymbol{\gamma}^{(-j,t)} - n\tau \mathbf{P}^{-1}(\mathbf{Z}'\mathbf{s} + 2\lambda_l \boldsymbol{\gamma}^{(-j,t)}),$$

for $t = 0, 1, 2, \ldots$ until convergence, where

$$\mathbf{P} = \mathbf{Z}'\mathbf{Z} + 2n\lambda_l \tau \mathbf{I}_m$$

and $\mathbf{s}$ is an $n$-vector whose $i$th entry is $\tilde{y}_i^{[j]} L'_\tau \left( \tilde{y}_i^{[j]} \mathbf{Z}' \boldsymbol{\gamma}^{(-j,t)} \right) / n$. We keep decreasing $\tau$ and repeat the above procedure until all the solutions satisfy the KKT conditions of problem (40).

In this algorithm, note the matrix inversion does not depend on $j$, so the computational cost is shared by all LOOCV computations.

## D    R Packages, Simulations, and Benchmark Data Sets

R packages:

1. ccvsvm:
   https://myweb.uiowa.edu/boxwang/index.html#software
2. magicsvm:
   https://myweb.uiowa.edu/boxwang/index.html#software
3. kernlab:
   https://cran.r-project.org/web/packages/kernlab/index.html
4. LIBSVM:
   https://cran.r-project.org/web/packages/e1071/index.html

Simulation code: https://anonymous.4open.science/r/2022-0764/

Data:

- arrhythmia:
  http://archive.ics.uci.edu/ml//datasets/Arrhythmia
- australian:
  http://archive.ics.uci.edu/ml/datasets/statlog+(australian+credit+
  approval)
- chess:
  https://archive.ics.uci.edu/ml/datasets/Chess+(King-Rook+vs.
  +King-Pawn)
- heart:
  https://archive.ics.uci.edu/ml/datasets/statlog+(heart)
- leuk:
  https://rdrr.io/cran/MASS/man/leuk.html
- malaria:
  https://www.nature.com/articles/npre.2011.5929.1.pdf?origin=ppub
- musk:
  https://archive.ics.uci.edu/ml/datasets/Musk+(Version+1)
- sonar:
  https://archive.ics.uci.edu/ml/datasets/Connectionist+Bench+(Sonar,
  +Mines+vs.+Rocks)
- valley:
  http://archive.ics.uci.edu/ml/datasets/hill-valley