# OpenReview forum: "A Consolidated Cross-Validation Algorithm for Support Vector Machines via Data Reduction"
_NeurIPS.cc/2022/Conference — NeurIPS 2022 Accept_

### Official Review · Reviewer_Tqdk · 2022-07-10

**Rating:** 8
**Confidence:** 4
**Soundness:** 4 excellent
**Presentation:** 4 excellent
**Contribution:** 4 excellent

**Summary:**

This paper presents a data reduction strategy and an algorithm that learns an SVM and a leave-one-out-cross-validated version much faster than the standard scheme which would require training once for every fold. The paper exploits the sparsity within the solution process to reduce the amount of computation required. The paper describes the theory around this and several experiments showing greatly superior running times from the scheme.

**Questions:**

1. Do the authors have some sense of how much computation the data reduction strategy saves and the characteristics of the problem that relate the most with computation reduction? This is in line with what I listed as the second weakness of the paper.

**Limitations:**

The authors clearly addressed the limitations of their work. There is no potential negative societal impact of their work that they can account for and affect.

**Strengths And Weaknesses:**

Strengths:
1. The development is clearly an important development for SVMs, which are an important class of algorithms. It represents an original and clever development.
2. The paper has convincing experimental results, showing that the new algorithm achieves nearly the same objective values but in much lower running time.

Weaknesses:
1. The paper should provide the algorithm in pseudocode form to ease reproducibility due to the multiple parts of the algorithm. If space is a concern, then the key optimization problems or equations can be put in the pseudocode and then cited in the text.
2. There should be more analysis of how effective the data reduction strategy suggested by equation 5 is. Clearly it is effective as demonstrated by the running times. However, I think the authors should still show what fraction of computations is eliminated due to the strategy and give a sense of the factors in the dataset that are most responsible for reduction in computations.

---

> ### Author Response · Authors · 2022-08-02
> **Response to Reviewer Tqdk**
>
> Thank you for all your comments.
>
> > The paper should provide the algorithm in pseudocode form to ease reproducibility due to the multiple parts of the algorithm. If space is a concern, then the key optimization problems or equations can be put in the pseudocode and then cited in the text.
>
> **A**: We have added the pseudocode in Section B of the supplemental materials to introduce the consolidated CV algorithm developed in Section 2 and the general SVM problems in Section 3. We will put these back to the main paper when an extra page becomes available in the camera-ready.
>
> >  There should be more analysis of how effective the data reduction strategy suggested by equation 5 is. Clearly it is effective as demonstrated by the running times. However, I think the authors should still show what fraction of computations is eliminated due to the strategy and give a sense of the factors in the dataset that are most responsible for reduction in computations.
>
> **A**:
> In the revised paper, we provide an example in the simulation to study the computation time of _magicsvm_ and _ccvsvm_ and investigate how _ccvsvm_ speeds up the algorithm exactly. We observe that
> - In _magicsvm_, most computational efforts (about 95\%) are made for computing the LOOCV solutions, i.e., solving problem (3) in the paper, and the remaining time is mainly for matrix inversions;
> - In contrast, _ccvsvm_ only needs half of that time for the LOOCV computation. Because _ccvsvm_ transforms problem (3) into problem (9), according to equation (5), _ccvsvm_ can more efficiently perform LOOCV by solving optimization problems with reduced dimensions.

---

### Official Review · Reviewer_uBSd · 2022-07-11

**Rating:** 7
**Confidence:** 4
**Soundness:** 3 good
**Presentation:** 3 good
**Contribution:** 3 good

**Summary:**

This paper proposed a computationally efficient algorithm to perform leave-one-out cross-validation using the support vector machine with Gaussian kernel. The authors proposed a two-stage cross-validation that is applicable to the support vector machine with the bias term.

**Questions:**

Major questions:
* For a support vector machine classifier, while training if we remove a data point that is not a support vector and add a data point that is not a support vector from the support vector machine trained using the full data, the solution does not change. Is this property used in the implementation?
* Can the proposed algorithm be applied to support vector regression algorithm with the $\epsilon$-insensitive error function?
* Can the proposed algorithm be applied to arbitrary kernels with unbounded values such as polynomial kernels?

Minor questions:
* Is there a typo in the equation just after line 109?

**Limitations:**

Yes.

**Strengths And Weaknesses:**

Strengths:
* Nice and detailed theoretical analysis.

Weaknesses:
* Empirical results on small data sets.

---

> ### Author Response · Authors · 2022-08-02
> **Response to Reviewer uBSd**
>
> Thank you for all your comments.
>
> > For a support vector machine classifier, while training if we remove a data point that is not a support vector and add a data point that is not a support vector from the support vector machine trained using the full data, the solution does not change. Is this property used in the implementation?
>
> **A**: Yes, this property is featured in _ccvsvm_. To clarify this, we added pseudocode in the revised paper to illustrate this acceleration strategy by avoiding redundant computation on non-support-vector points.  Thanks for this insightful comment.
>
>
> > Can the proposed algorithm be applied to support vector regression algorithm with the $\epsilon$-insensitive error function?
>
> **A**: Although _ccvsvm_ cannot be directly applied to SVR (since our work is based on the exact leave-one-out formula only for classification problems so far), the idea of consolidated data reduction is general and extensible. We can extend it to SVR and other regression problems like quantile regression for solving the solution paths and performing data reduction in a similar consolidated way. We briefly discussed this interesting direction in the discussion section and hope to conduct investigations for future work.
>
>
> > Can the proposed algorithm be applied to arbitrary kernels with unbounded values such as polynomial kernels?
>
> **A**: The proposed algorithm can be applied to arbitrary kernels. Although some kernels might be unbounded, e.g. polynomial kernels, the term $B = \max_i K(x_i, x_i)$ involved in the computation depends only on the training data, thus $B$ can be calculated as long as the data are realized. Also, since the data are typically standardized, $B$ is not extremely large in practice. We clarified this point under Theorem 2.2 in the revised paper.
>
>
> > Is there a typo in the equation just after line 109?
>
> **A**: Apologies for the typo: $\partial L(t)$ should be 1, not -1.

---

### Official Review · Reviewer_vgth · 2022-07-11

**Rating:** 3
**Confidence:** 4
**Soundness:** 2 fair
**Presentation:** 2 fair
**Contribution:** 2 fair

**Summary:**

In this paper, an integrated Cross-Validation procedure is introduced for one of the leading classifiers, kernel SVM, and an algorithm called ccvsvm is developed. This algorithm is based on the recently proposed leave-one-out lemma and magicsvm algorithms. Numerical experiments demonstrate that the algorithm is more than an order of magnitude faster than the two SVM solvers but with almost the same accuracy compared to existing data reduction methods.

**Questions:**

Q1: Numerical experiments demonstrating the effectiveness of the proposed method have been carried out with a maximum number of cases of around 3,000. Would the proposed method still be effective if, for example, the number of instances exceeds 10,000?

Q2: Do you already have ideas for extending the proposed method to machine learning algorithms other than SVM? If so, it would improve the value of the paper if you could present them in the Discussion.

**Limitations:**

This paper is not applicable as there are no problems in terms of limitations or potential societal impact.

**Strengths And Weaknesses:**

Strengths:
- The proposed method can perform leave-one-out cross-validation very fast, as shown by theoretical aspects and numerical experiments.

- Demonstrations are carried out using kernlab and LIVSVM, the main packages of existing support vector machines.

Weaknesses:
- Due to its reliance on support vector machine mechanisms, it is currently impossible to demonstrate its effectiveness with other machine learning algorithms.

- Numerical experiments have been carried out on relatively small benchmark data sets. For example, the performance has not been tested when the number of cases exceeds 10 000.

---

> ### Author Response · Authors · 2022-08-02
> **Response to Reviewer vgth**
>
> Thank you for your comments.
>
> > Due to its reliance on support vector machine mechanisms, it is currently impossible to demonstrate its effectiveness with other machine learning algorithms; Do you already have ideas for extending the proposed method to machine learning algorithms other than SVM? If so, it would improve the value of the paper if you could present them in the Discussion.
>
> **A**: Our methods  focus on pursuing the maximum algorithm speedup by utilizing the special structure of the kernel SVM. It has high impact as the kernel SVM is one of the most popular nonlinear learning algorithms with wide practical applications. It is especially effective in high-dimensional cases where the number of dimensions is greater than the number of samples.
>
> Although the ccvsvm algorithm is tailored for the SVM, its core idea is general and can be extended to other methods. For example, we can generalize the idea to the group lasso penalized logistic regression to speed up the computation of its LOOCV. Originally, the penalized logistic regression model can be typically computed using group-wise coordinate descent with a data reduction strategy that screens inactive groups before each group-wise update. Usually, determination of the step size during each update can be very expensive, and the computation needs to be repeated on each fold. Instead, we can adopt the ccvsvm idea to consolidate the computation of step sizes. After using the magic CV formula to force all the cross-validated data to have the same predictors, we then apply data reduction to screen out the redundant groups for all the cross-validated data, rather than reducing data individually in each fold. The same strategy can be extended to other classifiers with sparse penalties.
>
> The idea of consolidated data reduction can be generalized for regression methods when solutions are computed with a grid of tuning parameters. Rather than applying data reduction for an individual tuning parameter, we can apply it in a consolidated way for a group of consecutive tuning parameters. This approach gives identical reduced kernel matrices for the group of tuning parameters and thus saves time by avoiding the computation of different kernel matrices. Many data reduction methods may have the potential to be performed in a consolidated fashion.
>
> >   Numerical experiments have been carried out on relatively small benchmark data sets. For example, the performance has not been tested when the number of cases exceeds 10,000. Numerical experiments demonstrating the effectiveness of the proposed method have been carried out with a maximum number of cases of around 3,000. Would the proposed method still be effective if, for example, the number of instances exceeds 10,000?
>
> **A**: Our study mainly focuses on the exact SVM solutions for small and moderate data analysis, as there are wide practical applications with data of such sizes, and the kernel SVM is one of the most successful classifiers for those applications.
>
> However, this does not necessarily mean that our method cannot work with large-scale data. For such scenario, we suggest incorporating kernel approximation into the existing consolidated CV algorithm. Specifically, random features (Rahimi and Recht, 2007)  or Nyström subsampling (Rudi et al., 2015)
> can be applied in the exact leave-one-out formula of the SVM to find a low-cost approximation of the kernel matrix. Integrating these approximation techniques into our methods can further improve the numerical performance. These strategies can also improve generalization performances as they induce a form of implicit _computational_
> _regularization_. In the discussion section and Part C of the supplemental materials, we develop consolidated CV methods with kernel approximation, essentially converting the original consolidated kernel SVM to a consolidated linear SVM, which then can be efficiently solved by the proposed _ccvsvm_ algorithm. To give a quick demonstration, we consider the simulation example in Section 4.1 with $p = 20$ and increase $n$ to be $5,000,000$. Averaged by 50 runs, the SVM with random features can be rapidly trained and tuned in $831$ seconds, giving test error $0.286$ which is close to Bayes error, $0.260$. The computation time is only $15.7$ seconds when $n=100,000$. However, when $n$ is $800$, the test error of the SVM with random features is $0.351$, which is well above $0.309$, the test error of our exact kernel SVM solver given in Table 1. We leave full investigations of this strategy to future works.

---

### Official Review · Reviewer_5kDp · 2022-07-11

**Rating:** 7
**Confidence:** 5
**Soundness:** 4 excellent
**Presentation:** 3 good
**Contribution:** 3 good

**Summary:**

The authors propose a consolidated crossed validation for Support Vector Mahines (ccvsvm), based on the recent leave-one-out lemma and magicsvm, having as objective to reduce the training time via data reduction. Specifically, this two-stage ccvsvm deals with the SVM bias term, which is not handled by some existing data reduction methods. The authors demonstrate with numerical experiments that the proposed algorithm is about an order of magnitude faster than the two mainstream SVM solvers, kernlab and  LIBSVM, with almost the same accuracy.

**Questions:**

•	The manuscript section organization is missing from the introduction, it should be included.
•	One particular characteristic of this proposed solution is the sparsity on the solution of SVM. Is it possible to use the solution of the k-1 fold to improve the initial solution on the k-fold (finding the initial support vectors and non-zero alpha values)?


**Limitations:**

The authors address some of the method’s limitations and provide some future lines to generalize to other K-fold CV or other computationally-expensive ML methods

**Strengths And Weaknesses:**

•	Strengths: The core idea of the work is original in the sense of thinking at the same time data reduction (as part of CV), the QP optimization process and the LOO at the same time. The quality of the research work is solid. The work is clear and simple in explaining the state of the art, their contribution, the definition of their methods, and their general experimental setting.
•	Weaknesses: It will be great to have the code available to (at least partially) reproduce the experiments. Section 4.1 is of particular interest and table 1 underline how the proposed ccvsvm algorithm results efficient, but, again, reproducibility is a key. This comment also applies to section 4.2.

---

> ### Author Response · Authors · 2022-08-02
> **Response to Reviewer 5kDp**
>
> Thank you for all your comments.
>
> > It will be great to have the code available to (at least partially) reproduce the experiments. Section 4.1 is of particular interest and table 1 underline how the proposed _ccvsvm_ algorithm results efficient, but, again, reproducibility is a key. This comment also applies to section 4.2.
>
> **A**: In the spirit of reproducibility, we created an R package for _ccvsvm_ and provided code for reproducing simulation results and real data analyses in the paper. The download links are provided in Section D of the supplementary materials. For readers' convenience, we also provided pseudocode of our algorithms in Section B of the supplementary materials.
>
> > The manuscript section organization is missing from the introduction, it should be included.
>
> **A**: Thank you for the suggestion. The section organization is now included in the revised paper.
>
> > One particular characteristic of this proposed solution is the sparsity on the solution of SVM. Is it possible to use the solution of the k-1 fold to improve the initial solution on the k-fold (finding the initial support vectors and non-zero alpha values)?
>
> **A**: Thanks for this insightful comment. As mentioned in Section 2.3, we included the warm-start strategy and Nesterov's acceleration into _ccsvm_ to further speed up the computation. In particular, for each tuning parameter $\lambda_l$, we used the solution from the complete problem $\hat{\boldsymbol{\eta}}_l$, as the initial value to solve the LOOCV problems $\hat{\boldsymbol{\eta}}_l^{[-j]}$ in problem (8), leading to a "warm-start" algorithm. Our approach essentially shares the same spirit as the method you suggested. In the revised paper, we added some more comments to illustrate this warm-start strategy and also included it in pseudocode.

---

### Author Response · Authors · 2022-08-09
**Acknowledgement**

As the author-reviewer discussion period is coming to an end, we would like to thank all the reviewers for their comments and valuable time reviewing our paper, which helped us improve the quality of the paper and discuss the potential extensions. We hope the concerns have been addressed in our response. If there are more questions, we'd be happy to clarify and discuss.

Thank you very much for your time and consideration! Much appreciated!

---

### Meta-Review · Area_Chair_t3ye · 2022-08-26

**Recommendation:** Accept
**Confidence:** Certain

**Metareview:**

The paper proposes a new algorithm for training kernel support vector machines (SVM), that exploits a leave one out lemma to develop a significantly faster algorithm. The key advances are a data reduction approach, a consolidated cross validation method, and a warm start. Empirical comparisons to magicsvm, kernlab, and LIBSVM demonstrate an order of magnitude speedup. The paper proposes a major advance to kernel SVMs, without resorting to approximations or linearization. The combination of theoretical insight, exploitation of the structure of the method, as well as careful reuse of computation are all excellent contributions.

Four reviewers considered the submission, and three reviewers were very pleased by the novel approach for a classic algorithm. One reviewer found several issues with the submission, and did not further engage during the rebuttal period. A different reviewer actually was positively influenced by the author response to the negative review.

It gives me great pleasure to recommend this paper for acceptance to NeurIPS 2022. Congratulations!

**Award:**

No

---

### Decision · Program_Chairs · 2022-09-14

Accept